# GABAergic inhibition in dual-transmission cholinergic and GABAergic striatal interneurons is abolished in Parkinson disease

N. Lozovaya[1], S. Eftekhari[2], R. Cloarec[2], L.A. Gouty-Colomer[2], A. Dufour[2], B. Riffault[2], M. Billon-Grand[2], A. Pons-Bennaceur[3], N. Oumar[1], N. Burnashev[3], Y. Ben-Ari[1,2] & C. Hammond[1,3]

We report that half striatal cholinergic interneurons are dual transmitter cholinergic and GABAergic interneurons (CGINs) expressing ChAT, GAD65, Lhx7, and Lhx6 mRNAs, labeled with GAD and VGAT, generating monosynaptic dual cholinergic/GABAergic currents and an inhibitory pause response. Dopamine deprivation increases CGINs ongoing activity and abolishes GABAergic inhibition including the cortico-striatal pause because of high $[Cl^-]_i$ levels. Dopamine deprivation also dramatically increases CGINs dendritic arbors and monosynaptic interconnections probability, suggesting the formation of a dense CGINs network. The NKCC1 chloride importer antagonist bumetanide, which reduces $[Cl^-]_i$ levels, restores GABAergic inhibition, the cortico-striatal pause-rebound response, and attenuates motor effects of dopamine deprivation. Therefore, most of the striatal cholinergic excitatory drive is balanced by a concomitant powerful GABAergic inhibition that is impaired by dopamine deprivation. The attenuation by bumetanide of cardinal features of Parkinson's disease paves the way to a novel therapeutic strategy based on a restoration of low $[Cl^-]_i$ levels and GABAergic inhibition.

[1] B&A Therapeutics, Ben-Ari Institute of Neuroarcheology, Batiment Beret-Delaage, zone Luminy entreprises, 13288 Marseille, Cedex 09, France.
[2] Neurochlore, Ben-Ari Institute of Neuroarcheology, Batiment Beret-Delaage, Zone Luminy Biotech Entreprises, 13288 Marseille, Cedex 09, France. [3] INMED, INSERM U901 and Aix-Marseille University, 13273 Marseille, Cedex 09, France. These authors contributed equally: Y. Ben-Ari, C. Hammond. These authors jointly supervised this work: Y. Ben-Ari, C. Hammond. Correspondence and requests for materials should be addressed to Y.B-A. (email: ben-ari@batherapeutics.com)

Cholinergic interneurons are key modulators of striatal microcircuit function and instrumental in the processing and reinforcement of goal-directed and habitual reward-related behaviors[1–6]. Cholinergic interneurons that are the only intrinsic excitatory neurons in the striatum, fire regularly[2,5–8]. They are also the biggest striatal neurons with well-characterized intrinsic electrical features. Cholinergic interneurons ramify extensively within the striatum controlling the activity of GABAergic interneurons and spiny projection neurons (SPNs), the latter constituting the dominant striatal neuronal type and sole striatal output[8–10]. Cortical or thalamic stimulation generates a characteristic pause response in cholinergic interneurons that has been linked to motor learning[1,2,5,6,8]. Other observations suggest that thalamic inputs to striatal cholinergic neurons provide a neural substrate for attentional shifts and cessation of motor activity in response to salient stimuli[1]. Collectively, these observations stress the nodal role of cholinergic neurons in motor integrative activity. In keeping with this, in dopamine-deprived conditions, cholinergic interneurons are hyperactive, generating exacerbated oscillations that underlie many clinical manifestations and deleterious sequels of Parkinson's disease (PD)[11,12].

Cholinergic interneurons are considered as a homogeneous population in their morphological and electrical properties differing from other non-cholinergic striatal neurons. However, genetic fate mapping studies suggest some degree of heterogeneity within the striatal cholinergic population. Thus, adult cholinergic interneurons are generated from the medial ganglionic eminence expressing LIM homeobox 7 (Lhx7) but not Lhx6 in contrast to other striatal GABAergic interneurons[13,14]. However, a subset of cholinergic neurons is derived from Lhx6-expressing precursors and the remaining from precursors that have never expressed Lhx6[15]. In addition, medial septal origin has been recently demonstrated[16] reflecting the complexity of striatal cholinergic networks.

We now report that half the striatal cholinergic interneurons are in fact dual transmitter neurons, releasing acetylholine (ACh) and γ-aminobutyric acid (GABA), and generating mixed monosynaptic excitatory/inhibitory postsynaptic currents (PSCs) in their targets. These dual cholinergic/GABAergic interneurons (CGINs) have different electrical and morphological features from other cholinergic interneurons (CINs). The cholinergic excitatory/GABAergic inhibitory balance is perturbed by dopamine deprivation in CGINs with a failure of inhibition due to high $[Cl^-]_i$ levels. In these conditions, CGINs are overactive with a dramatic increase of monosynaptic CGIN–CGIN interconnections. The NKCC1 chloride importer antagonist bumetanide, which restores low $[Cl^-]_i$ levels, attenuates electrophysiological and motor manifestations of dopamine deprivation, stressing the importance of CGINs $[Cl^-]_i$ levels, and GABAergic inhibition in PD and paving the way to a novel non-dopaminergic treatment of the disease.

## Results
### CINs and CGINs are two cholinergic interneuron populations.
Using Lhx6-iCre;RCE-EGFP mice, we first investigated the properties of EGFP$^+$ cholinergic interneurons generated from Lhx6-expressing precursors[17,18]. We identified a subpopulation of EGFP$^+$ interneurons that were labeled by ChAT and Lhx6 (Fig. 1a and Supplementary Fig. 1a). ChAT and Lhx6 co-immunolabeling was also observed in wild-type (wt) mice (Supplementary Fig. 1b) validating the presence of both Lhx6-positive and -negative cholinergic interneurons. These observations raised the possibility that ChAT- and Lhx6-positive interneurons might be cholinergic and GABAergic[19]. To test that hypothesis, we performed single cell reverse transcription quantitative PCR (RT-

qPCR) analysis. This revealed that EGFP$^+$ neurons endowed with large soma (30–50 μm), firing, and morphological properties typical for cholinergic interneurons (Fig. 1b) expressed ChAT and Lhx7 messenger RNAs, but also GAD 65 (90%, $n = 10/11$) and Lhx6 mRNAs (46%, $n = 5/11$) (Fig. 1c and Supplementary Fig. 2), suggesting that these interneurons have indeed GABAergic markers. We shall respectively refer to EGFP$^+$ and EGFP$^-$ interneurons as CGINs and CINs.

We then compared the intrinsic and morphological features of CINs and CGINs. They shared several major common intrinsic membrane properties but differed in that CGINs had higher sag amplitudes in response to hyperpolarizing pulses than CINs, lower spontaneous spiking frequency, and smaller dendritic arbors (Fig. 1b, d–h, Supplementary Fig. 3, and Supplementary Tables 1-3). CINs and CGINs differed from other non-cholinergic EGFP$^+$ interneurons, notably fast spiking (FS), and persistent and low-threshold spike (PLTS) interneurons, in morphological and electrical features including ongoing spike frequency (Fig. 1b, f–h, Supplementary Fig. 3, and Supplementary Tables 2, 3). Therefore, CGINs and CINs have different features but share common properties that differentiate them from other non-cholinergic interneurons.

**Strong inhibitory pause response in CGINs.** The cortically evoked pause and rebound response of cholinergic interneurons is considered an important element in cortico-striatal interactions and targeted motor behavior[2,5,6,11]. We first tested if cortical stimulation generates a similar response in CGINs. Extracellular cortical stimulation (5 stimuli, 50 Hz train) evoked in CGINs large glutamatergic PSCs followed by nicotinic and GABAergic PSCs, reflecting the complexity of the striatal network response to glutamatergic inputs (Supplementary Fig. 4). In non-invasive cell-attached (ca) recordings, the stimulation triggered a burst followed by a pause and a rebound increase of spontaneous spiking frequency (Fig. 1i, k, m and Supplementary Table 4). In contrast, this stimulation generated in CINs a significantly smaller and shorter duration inhibitory pause and no rebound (Fig. 1j, l, m). Interestingly, focal applications of the GABA$_A$ receptor agonist isoguvacine inhibited ongoing activity in CGINs but not in CINs (Supplementary Fig. 5 and Supplementary Table 5), suggesting stronger GABAergic signaling in the former than the latter. Therefore, the pause response and GABAergic inhibition are more prominent in CGINs than in CINs stressing their differences and suggesting a more prominent role of the GABAergic drive in CGINs than in CINs.

**GABAergic markers expressed by CGINs.** To further confirm the GABAergic phenotype of CGINS, we examined whether cholinergic neurons express GAD67 and/or GAD65 by co-immunostaining for ChAT and green fluorescent protein (GFP) in dorso-lateral striatum slices from Gad1-GFP mice. We detected co-labelling of GAD65/67 and ChAT but no co-labeling of ChAT and EGFP, suggesting that ChAT$^+$ neurons might be immunopositive for GAD65, but not for GAD67 (Fig. 2a and Supplementary Fig. 6a). This was validated in wt mice where the same overlap between ChAT and GAD 65/67 was observed (Supplementary Fig. 6b). To quantify CGINs, we shifted to vesicular GABA transporter (VGAT) antibodies that have been extensively used to label GABAergic neurons[20]: 59% of ChAT$^+$ neurons were also labeled with VGAT in Lhx6-iCre;RCE-EGFP and wt mice (Fig. 2b, Supplementary Fig 6c, and Supplementary Table 6), indicating that they have the molecular machinery for GABA release. Importantly, more than 95% of these ChAT$^+$ VGAT$^+$ cells were also positive for EGFP in Lhx6-iCre;RCE-

EGFP mice, validating Lhx6 as a marker for GABAergic features of CGINs (Supplementary Table 6). Collectively, these observations suggest that half of cholinergic striatal interneurons have the ability to synthesise and release ACh and GABA.

To determine the number of CGINs and CINs in the entire dorsolateral striatum, we used the immunolabeling-enabled three-dimensional imaging of solvent-cleared organs (iDISCO)

clarification technique in Lhx6-iCre;RCE-EGFP mice[21,22]. We found that 53% of all ChAT[+] interneurons ($n = 5617$ cells) were EGFP-positive (CGINs, $n = 2933$ cells) and 47% EGFP-negative (CINs, $n = 2684$ cells) (Fig. 2c, Supplementary Table 7, and Supplementary Movie 1). Therefore, the CGIN network and its dual transmission might have a nodal role in the operation of striatal networks.

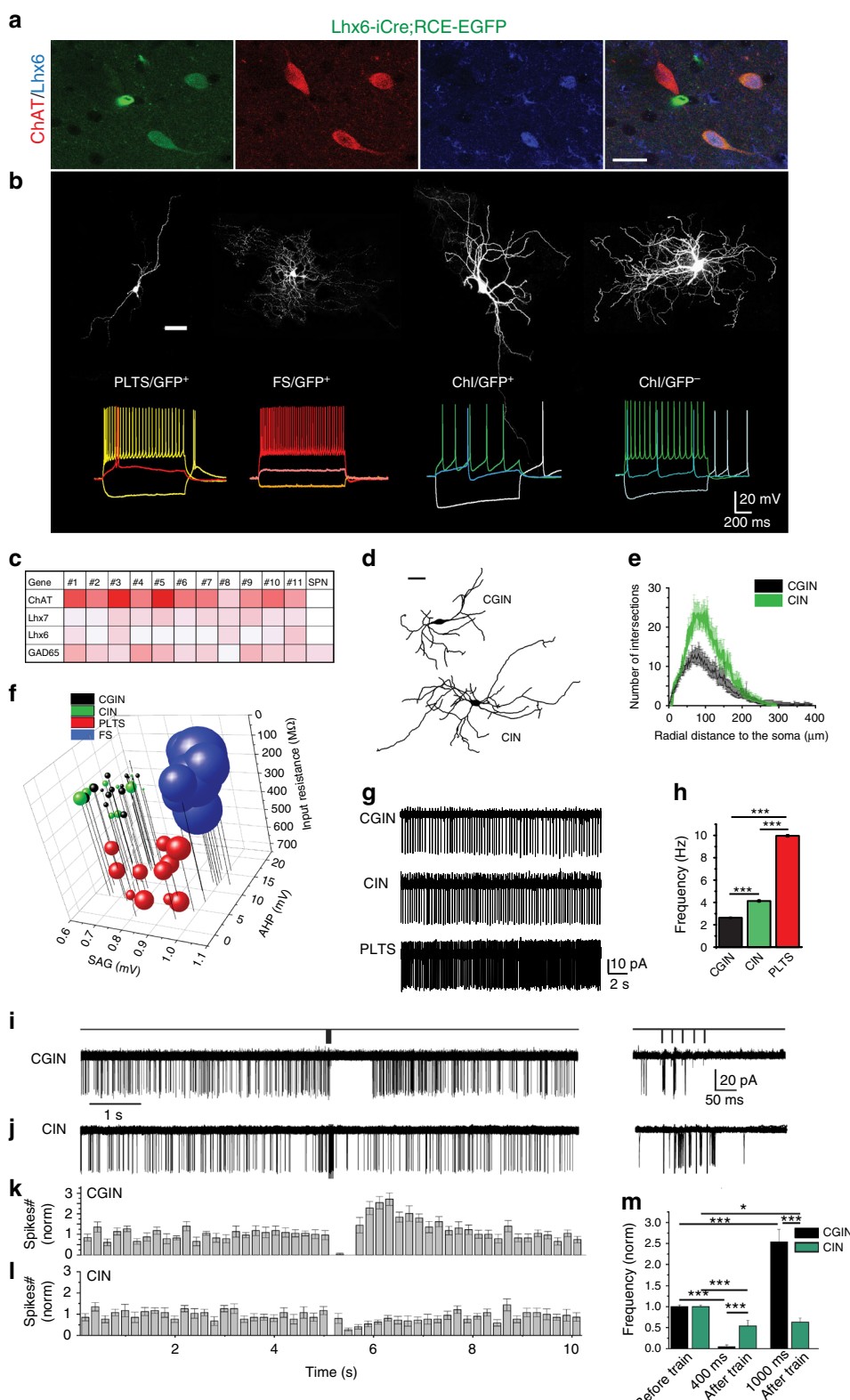

**Dual ACh and GABA PSCs generated by CGINs.** To test whether CGINs generate composite cholinergic and GABAergic PSCs, we used both ChAT-ChR2-EYFP and ChAT-ChR2-EYFPxLhx6-iCre; AI14-tomato mice to identify and light stimulate CGINs (yellow fluorescent protein (YFP)/tomato expression). In these mice, 98% or 95% of $EYFP^+$ neurons are $ChAT^+$, respectively, (Supplementary Fig. 7a and Supplementary Tables 8 and 9). Optogenetic stimulation of presynaptic CGINs with short (2–10 ms) focal light pulse generated a single action potential (AP) in $EYFP^+$ (channelrhodopsin-2 (ChR2) expressing) neurons (Supplementary Fig. 8). In whole-cell recordings, the same light stimulation generated in a postsynaptic CGIN an outward GABAergic PSC recorded at $V_H = 0$ mV and an inward PSC recorded at $V_H = -70$ mV (Fig. 3a–c). It evoked a direct monosynaptic GABAergic PSC (12%, 10 out of 86 pairs tested, pooled data obtained from six ChAT-ChR2-EYFP, and two ChAT-ChR2-EYFPxLhx6-iCre; AI14-tomato mice, Fig. 3a), a polysynaptic GABAergic PSC (26%, 22 out of 86 pairs tested, Fig. 3b), or a composite mono- and polysynaptic GABAergic PSC (Fig. 3c). A cocktail of nicotinic receptor antagonists blocked the polysynaptic but not the monosynaptic component of the GABAergic PSC (mean amplitude $39.7 \pm 8.0$ pA), the latter being fully blocked by the selective $GABA_A$ receptor antagonist gabazine (five pairs, pooled data obtained from six ChAT-ChR2-EYFP, and two ChAT-ChR2-EYFPxLhx6-iCre; AI14-tomato mice, Fig. 3c). The decay time constants of mono- and polysynaptic PSCs were clearly different ($30.5 \pm 0.3$ ms, $n = 5$ vs. $84.5 \pm 11.4$ ms, $n = 12$, $P < 6 \times 10^{-4}$, Fig. 3c). Optogenetic stimulation of presynaptic $ChAT^+$ cells in ChAT-ChR2-EYFP mice also induced monosynaptic GABAergic PSC in a postsynaptic $ChAT^+$ neuron (Supplementary Fig. 7a, b). In addition, in the presence of tetrodotoxin (1 μM) to abolish spontaneous activity and 4-AP (50 μM) to block $K^+$ currents and enhance release probability, optogenetic stimulation of a presynaptic $ChAT^+$ neuron generated a dual cholinergic /GABAergic PSC in a postsynaptic $ChAT^+$ neuron (Supplementary Fig. 9), demonstrating the dual release of GABA and acetylcholine by the terminals. Therefore, CGINs generate $GABA_A$ and ACh (nicotinic) receptor-mediated PSCs.

To investigate the properties of CGIN–CGIN connections, paired patch-clamp whole cell recordings of CGINs are essential. In pair CGINs recordings in Lhx6-iCre;RCE-EGFP mice, only one dual cholinergic/GABAergic monosynaptic CGIN-CGIN PSCs was found (3.5%; 1/29 pairs tested, Supplementary Fig. 10). Therefore, in control conditions, CGINs evoke dual cholinergic/GABAergic PSCs in other CGINs, albeit with a low probability in pair recordings.

**Dopamine deprivation strengthens CGIN–CGIN network.** GABAergic inhibition and notably chloride regulation are compromised in a wide range of pathological conditions leading to an imbalance between excitatory and inhibitory drives[23,24]. We

therefore investigated whether this is also the case for ACh/GABA co-transmission in mice that were dopamine-deprived by intra-striatal injections of 6-hydroxydopamine (6-OHDA) (Supplementary Fig. 11). Optogenetic techniques could not be used because of the high degree of lethality produced by 6-OHDA in ChAT-ChR2-EYFP mice ($n = 4/4$). We therefore performed pair recordings of CGIN–CGIN cholinergic/GABAergic PSCs in the 6-OHDA dopamine-deprived striatum of Lhx6-iCre; RCE-EGFP mice. A dual cholinergic/GABAergic PSC was observed in 22% of paired CGIN–CGIN recordings (4/18 tested pairs) (Fig. 3d, e). Outward GABAergic and inward nicotinic currents were monosynaptic with similar delays ($11.3 \pm 0.7$ ms and $11.5 \pm 0.8$ ms, respectively, $n = 4$ pairs, $P = 0.12$) (Fig. 3f). The synaptic delay of CGIN–CGIN interconnections was similar to the delay of nicotinic response evoked in CGINs by cortical stimulation, reflecting recurrent activation in CGIN network (see Supplementary Fig. 4c, d). The nicotinic and GABAergic identities of these monosynaptic PSCs was confirmed by their selective blockade by nicotinic and GABAergic receptor antagonists, respectively (Fig. 3g). Therefore, dopamine deprivation increases about seven times the probability of monosynaptic CGIN–CGIN connections.

This enhanced monosynaptic connectivity might be due to reactive sprouting produced by dopamine depletion. Post-hoc CGINs reconstruction revealed an exuberant increase of dendritic arbor (Fig. 3h, i and Supplementary Fig. 12), including the number of dendrites, dendritic nodes and tips, the total length and the critical value (Supplementary Table 1). The enlarged dendritic receptive field could facilitate connectivity by increasing the probability of CGIN–CGIN connections. Therefore, dopamine deprivation augments CGIN–CGIN interconnections.

**Dopamine deprivation abolishes GABAergic inhibition in CGINs.** The failure of GABAergic inhibition in pathological conditions often results from an impairment of chloride cotransporters leading to high $[Cl^-]_i$ levels[23,24]. We tested whether GABAergic inhibition is affected in a PD mouse model. Using non-invasive ca single-channel GABA current recordings in CGINs, the driving force for GABA currents ($DF_{GABA}$) was found more elevated in 6-OHDA-treated than in control striatum (see Fig. 4a–f and Supplementary Tables 10 and 11). In line with this, isoguvacine inhibited the spontaneous spike frequency of CGINs in control, but not in 6-OHDA-treated striatum, suggesting a failure of the inhibitory drive (Fig. 4g, h, j, Supplementary Fig. 13a, and Supplementary Tables 12 and 13). Bath application of bumetanide, the specific antagonist of the chloride importer NKCC1 that reduces $[Cl^-]_i$ levels[23], restored isoguvacine inhibition (Fig. 4i, j and Supplementary Table 12). The peak frequency distribution of CGIN spontaneous spiking was significantly shifted to higher values with a strong increase of mean frequency value in 6-OHDA-lesioned striatum versus control one (Fig. 4k, l, Supplementary Fig. 13b, and Supplementary Tables 14 and 15).

**Fig. 1** A subpopulation of cholinergic interneurons, CGINs, expresses cholinergic and GABAergic markers. **a** ChAT (red) and Lhx6 (blue) co-immunolabeling in $EGFP^+$ (green) cells in Lhx6-iCre;RCE-EGFP mice. **b** Top: Representative images of biocytin-filled PLTS, FS, $EGFP^+$, and $EGFP^-$ cholinergic interneurons (ChI) and (bottom) corresponding traces of their representative firing patterns. **c** RT-qPCR heat-map expression for ChAT, Lhx7, Lhx6, and GAD65 mRNAs in single cells (relative to HPRT mRNA; white to red: lowest to highest expression, SPN: negative control). **d** Representative reconstructed CIN and CGIN dendritic trees. **e** Sholl analysis of CGINs and CINs dendrites. **f** Distinct cluster formed by CGIN and CIN from PLTS and FS interneurons (3D plot of intrinsic properties). **g** Representative traces of cell-attached spontaneous spiking of CGIN, CIN, and PLTS interneuron. **h** Corresponding mean firing frequencies. **i**, **j** Top: Stimulation protocol; left: representative several superimposed consecutive **i** CGINs or **j** CINs responses to cortical train stimulation in cell-attached mode; right: responses within the train at extended time scale. Mean frequency histograms for **k** CGINs and **l** CINs. **m** Mean number of spikes during time windows after train stimulation (0–400 ms, 800–1000 ms) normalized to spikes counts before train. Scale bars: **a** 20 μm; **b**, **d** 50 μm. All means ± SEM. **e** Data sets were analyzed using Kolmogorov-Smirnov test; see Supplementary Table 1 for statistics. **h**, **m** Data sets were analyzed using one-way ANOVA followed by Fisher's least significant difference (LSD) post-hoc test; see Supplementary Table 2 for statistics; $*P < 0.05$, $***P < 0.001$

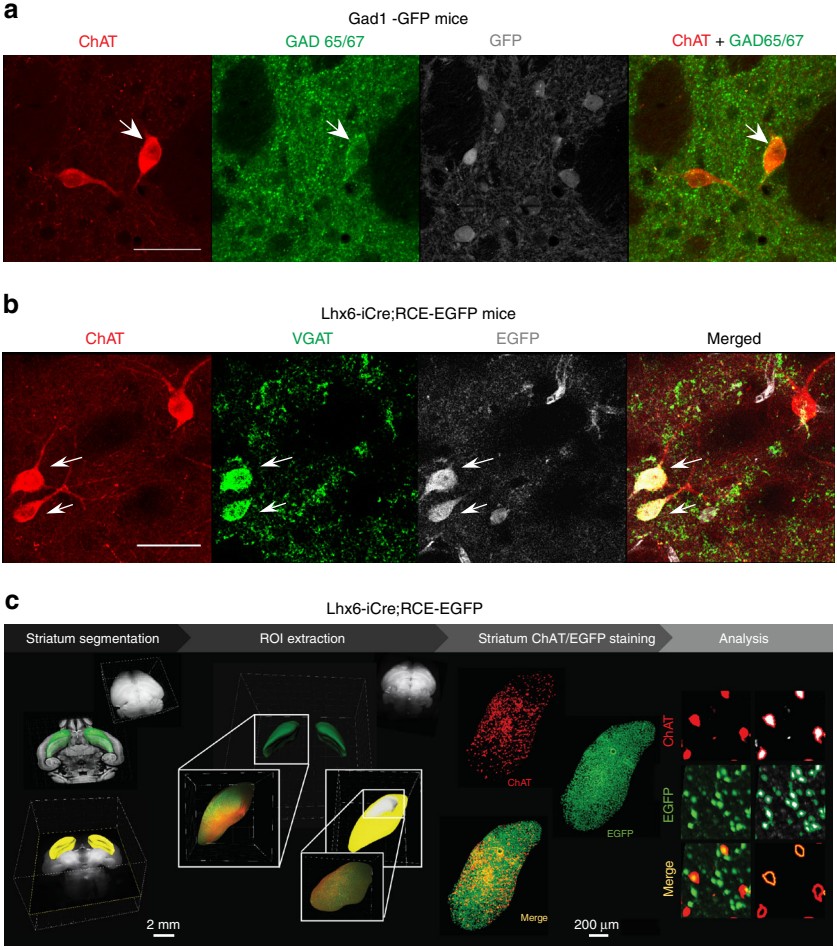

**Fig. 2** Identification and quantitative assessment of CGIN/CIN ratio in dorsolateral striatum using immunocytochemistry and iDISCO clarification techniques. **a** ChAT (red), GAD65/67 (green), and GAD1-GFP (gray) co-immunostaining of coronal striatal slices from GAD1-GFP mice showing expression of GAD65/67 in one of ChAT$^+$ cells (white arrow). Scale bar: 40 μm. **b** ChAT (red), VGAT (green), and EGFP (gray) co-immunostaining of coronal striatal slices from Lhx6-iCre;RCE-EGFP mice showing co-expression of VGAT and EGFP in a subset of ChAT$^+$ cells. Scale bar: 40 μm. See Supplementary Table 6 for statistics. **c** iDISCO experimental design: segmentation of the striatum, extraction of a region of interest (ROI, dorsolateral striatum) with IMARIS software. ChAT staining (red) in representative dorsolateral striatum from Lhx6-iCre;RCE-EGFP mice. Analysis, left column: immunofluorescence signal showing ChAT$^+$ and EGFP$^+$ Lhx6 cells; right column: cell profiler analysis with automatic cell detection (white) and cells co-immunopositive for ChAT and EGFP Lhx6 (orange) (see Supplementary Table 7 for statistics and Supplementary Movie 1)

Bumetanide significantly reduced spontaneous frequency in 6-OHDA-treated mice, restoring control ongoing frequency (Fig. 4k, l). Therefore, GABAergic inhibition is impaired in CGINs by dopamine deprivation because of high $[Cl^-]_i$ levels.

**Dopamine deprivation abolishes the CGIN pause response.** To determine the role of $[Cl^-]_i$ levels and GABAergic inhibition in control and dopamine deprived mice, we used non-invasive ca recordings of the pause-response in CGINs evoked by cortical stimulation. In control conditions, this triggered a burst of spikes followed by a pause (769.8 ± 74.3 ms, 6 cells, 3 mice) and a rebound increase of spontaneous spiking frequency (Fig. 5a, d, o, Supplementary Fig. 13c, and Supplementary Table 16). The stimulation evoked a similar pause-rebound response when neurons were recorded in whole cell current-clamp (cc) with low $[Cl^-]_i$ solution (Fig. 5b, e, p and Supplementary Table 17). The inhibition duration was identical in non-invasive ca and whole cell voltage clamp low $[Cl^-]_i$ recordings (at $V_H = V_R$) (Fig. 5a, c). In contrast, re-patching the same neurons in whole-cell configuration (at $V_H = V_R$) with a high $[Cl^-]_i$ solution abolished the pause

response and the rebound, and shifted the concomitant outward current to inward direction (Fig. 5f–h, p). The GABA receptor antagonist gabazine (10 μM) blocked the pause response (Supplementary Fig 14a, b and Supplementary Table 18). Therefore, $[Cl^-]_i$ levels determine the polarity of the GABAergic pause response.

A very different situation prevailed in 6-OHDA-treated mice; the pause-rebound was abolished in ca recordings (n = 7 cells, 3 mice). Instead, cortical stimulation increased spike frequency (Fig. 5i, k, o and Supplementary Table 16). Switching to whole cell current clamp in the same neuron with low $[Cl^-]_i$ solution restored the pause response (Fig. 5j, l, p). A similar restoration of the pause was produced by bumetanide. Thus, in ca recordings in 6-OHDA-treated mice, the pause response was restored (mean pause duration: 452.6 ± 41.9 ms, 8 cells, 3 mice), as well as the rebound spike frequency (Fig. 5m, n, o and Supplementary Table 16). Therefore, GABAergic inhibition in CGINs, which has a key role in the pause response, is abolished by dopamine deprivation because of high $[Cl^-]_i$ levels and rescued by reducing $[Cl^-]_i$ levels in patch recordings or by bumetanide administration.

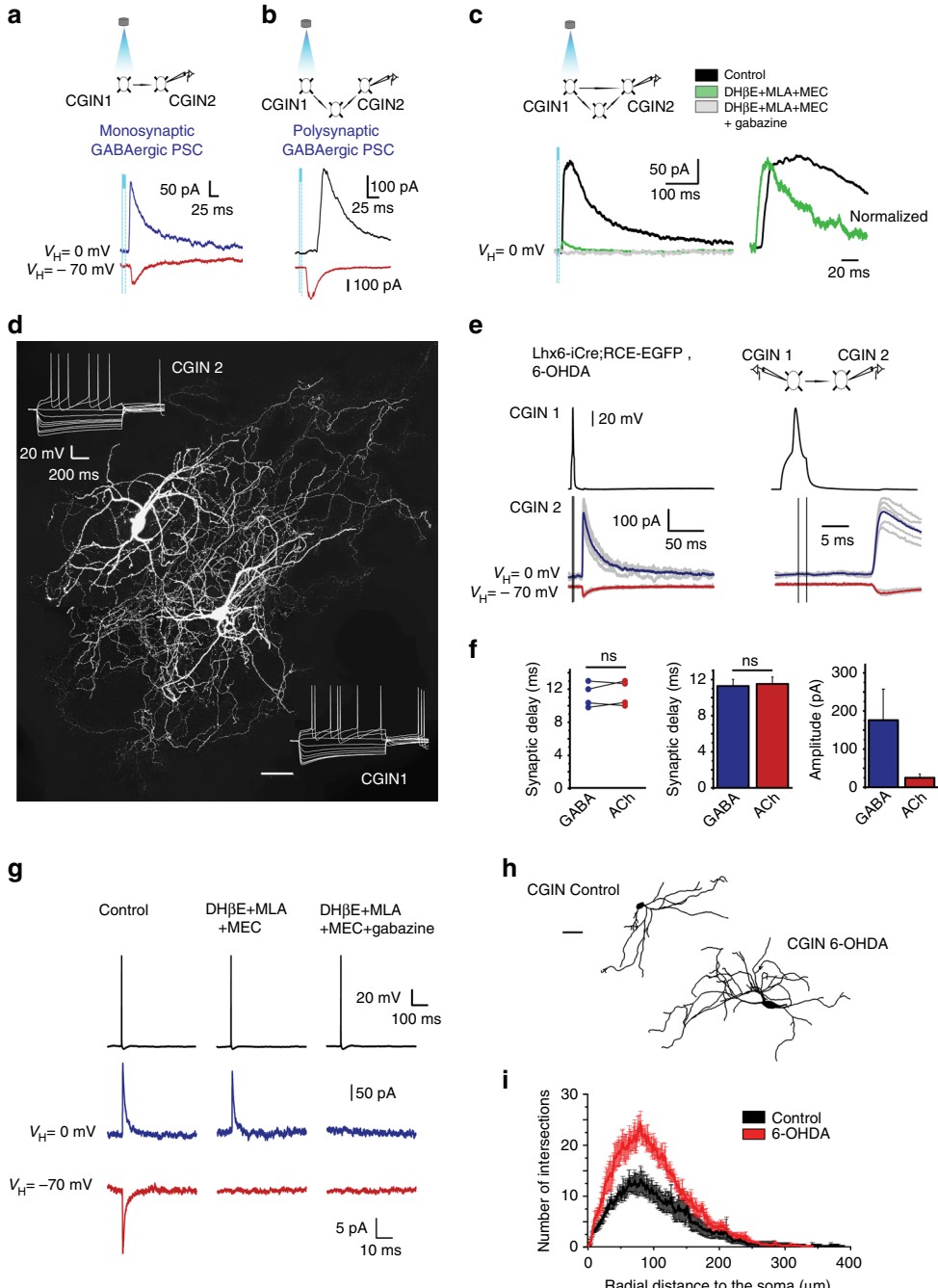

**Fig. 3** CGINs co-release ACh and GABA in control and 6-OHDA-treated mice. **a** Monosynaptic, **b** polysynaptic, or **c** mixed GABAergic PSCs evoked in postsynaptic whole cell recorded CGIN2 in response to optogenetic stimulation of presynaptic CGIN1 (control mice). **c** Left: Nicotinic receptor antagonists (MEC (10 μM), MLA (0.1 μM), and DHβE (10 μM)) blocked the polysynaptic but not the monosynaptic GABAergic PSC. Adding gabazine (5 μM) blocked the monosynaptic component. Right: Superimposed and normalized polysynaptic and monosynaptic GABAergic PSCs at extended scale. **d** Representative image of two identified biocytin-filled, monosynaptically-connected CGINs. **e** Left: Outward (GABAergic) and inward (cholinergic) PSCs evoked in postsynaptic CGIN2 (bottom) in response to a presynaptic spike generated in CGIN1 (top); right: same traces at an extended time scale. **f** Individual and averaged synaptic delays and amplitudes of GABAergic and cholinergic PSCs. **g** Blockade of inward and outward PSCs by nicotinic receptor antagonists and gabazine, respectively. **h** Representative reconstructed CGINs dendritic trees from control and 6-OHDA-treated mice. **i** Sholl analysis of CGINs dendrites from control and 6-OHDA-treated mice. Scale bar: **d** 50 μm and **h** 50 μm. All means ± SEM. **f** Significance was determined by two-tailed, unpaired Student's *t*-test; **i** data sets were analyzed using Kolmogorov-Smirnov test. See Supplementary Table 1 for statistics; NS, not significant

**Bumetanide attenuates effects of dopamine deprivation**. In earlier investigations, we showed that in dopamine-deprived conditions, SPNs that constitute over 95% of striatal neurons, generate spontaneous single or bursts of repetitive giant GABAergic currents (> 200 pA)[25]. These are relevant to PD being abolished by conditions known to attenuate PD symptoms including L-Dopa treatment or lesion of the subthalamic nucleus[26]. Applications of bumetanide or a cocktail of nicotinic

receptor antagonists (and not mecamylamine solely[25]), significantly decreased the frequency of giant GABAergic PSCs in SPNs (Fig. 6a–d and Supplementary Tables 19 and 20). Therefore, high $[Cl^-]_i$ levels and nicotinic receptor-mediated signals are instrumental in the generation of the giant GABAergic currents in SPNs in dopamine-deprived conditions.

To test whether motor symptoms of PD are also attenuated by bumetanide, we used the roller and pole behavior tests. 6-OHDA-lesioned mice showed severe deficits in both tests, these were rescued by chronic administration of bumetanide (Fig. 6e, f, Supplementary Movies 2-7, and Supplementary Tables 21 and 22). Therefore, high $[Cl^-]_i$ levels and GABAergic inhibition

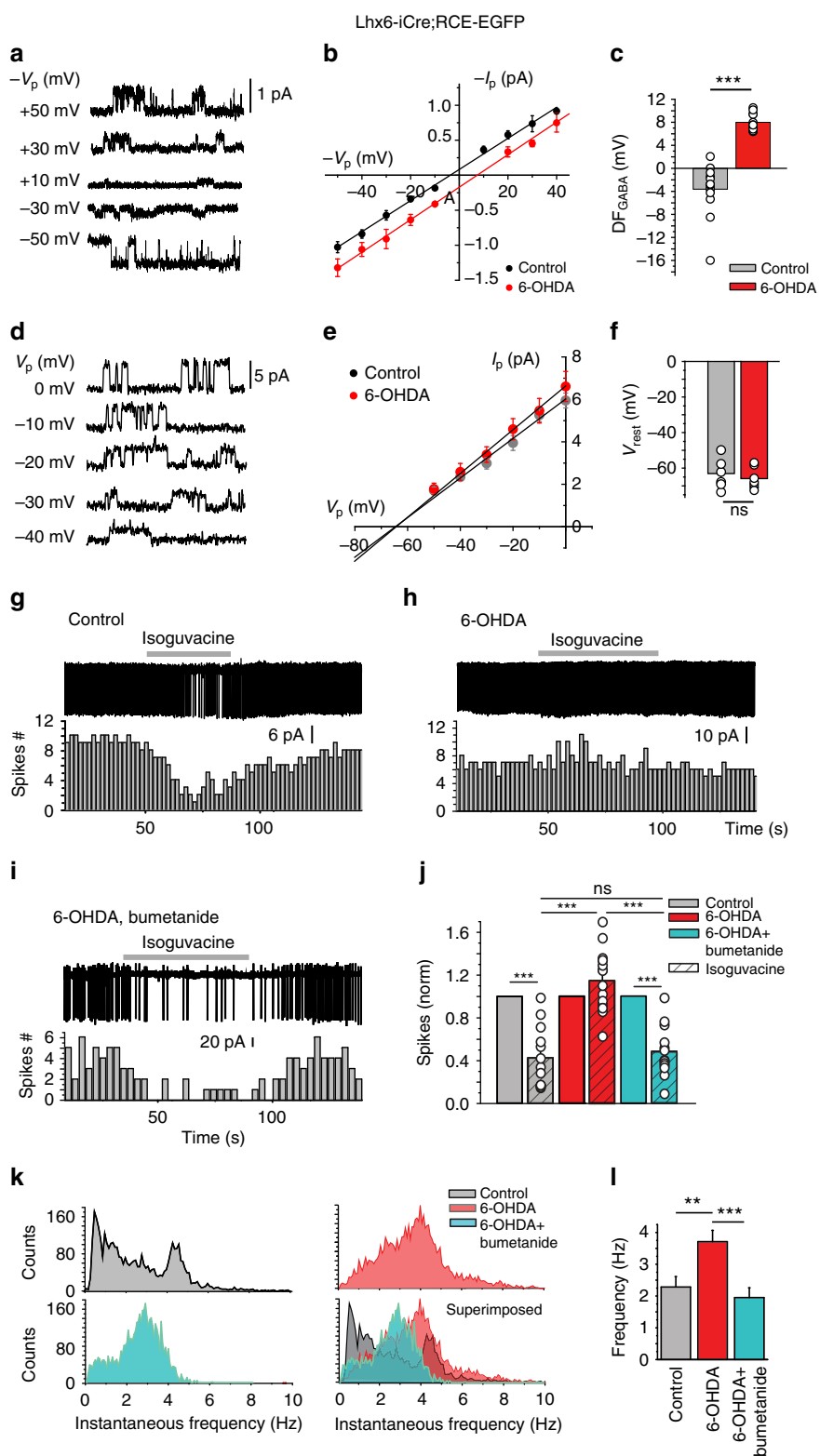

failure in CGINs contribute to motor deficiency produced by dopamine deprivation.

## Discussion

Co-release of transmitters has been observed in many neuronal systems and shown to modulate integrative functions. The co-released transmitters may control the balance between excitatory and inhibitory drives and exert similar or synergistic effects[27–32]. Cholinergic interneurons are the sole intrinsic excitatory neurons in the striatum whereas all other neuronal types are GABAergic. They are usually considered as a homogeneous population in their intrinsic, morphological and functional features. Here we show dual ACh and GABA co-transmission in half the striatal cholinergic interneurons (CGINs), relying on double immuno-histochemistry, single-cell RT-qPCR, iDISCO clarification technique, optogenetic stimulation and paired recordings. Although cholinergic and GABAergic PSCs have similar latencies, whether this co-transmission is due to co-release from the same terminals, different terminals or axonal branches remains to be determined. This dual transmission in CGINs will have to be incorporated in models of operation of the basal ganglia networks. This organization might have been evolutionary conserved linking basal ganglia to Central Pattern Generators[33]. However, the relative functional importance of these GABAergic inputs require further investigation.

The suggested operation of the dual ACh/GABA co-transmission is illustrated in a schematic diagram (Supplementary Fig. 15). Cholinergic interneurons directly innervate various types of GABAergic interneurons[5,34] and SPNs dendritic shafts or spines[8,9], raising the possibility of similar connections of CGINs. Similar to CINs, CGINs are autonomous pacemakers capable of maintaining spike activity in the absence of synaptic inputs[7]. In control conditions, interplay between dopamine innervation and intrinsic membrane properties, restrict ongoing activity, allowing the generation of behaviorally relevant oscillations and the pause response to sensory cues[2,35,36]. Whether the pause response is generated by the cortex or the thalamus as suggested by Surmeier and colleagues[1] cannot be ascertained at present, but our results are in full accord with this study in stressing the crucial role of CGINs in the synchronization of striatal networks in relation to sensory integration. Present results suggest that a balance between cholinergic excitation and GABAergic inhibition has an important role in this operation with intracellular chloride level in CGINs being a determinant factor.

In dopamine-deprived conditions, the excitation/inhibition equilibrium is impaired in CGINs because of high $[Cl^-]_i$ levels leading to the collapse of GABAergic inhibition and an unbalanced excitatory cholinergic drive. The additional contribution of a failure of inhibition in other interneurons—notably PLTS known to burst when dopamine deprived[25]—cannot be excluded. The pause that is correlated with a Go signal[37], disappears in dopamine-deprived

conditions[35]. The restoration of a pause by low $[Cl^-]_i$-containing pipettes or by bumetanide administration stresses the importance of CGINs $[Cl^-]_i$ levels in network dysfunction in PD models. There are several potential links between dopamine deprivation and failure of inhibition including the well-described inflammation produced by dopamine deprivation[38], which is associated with failure of inhibition[39]. Mixed cholinergic/Calretinin (presumably GABAergic) interneurons have been identified by immunocytochemistry in humans and non-human primates[40–42]. Interestingly, they are more prevalent in humans and primates than in rodents and their numbers increased in PD[40–42]. Therefore, the impairment of the GABAergic component in PD pathogenesis might be more important in primates than rodents. The amelioration of gait produced by bumetanide in a pilot open clinical trial[43] like that of pole test scores of 6-OHDA-treated mice (present results) suggest that chloride co-transporter-based agents might be useful in the treatment of PD symptoms.

## Methods

**Animals.** Hemizygous Lhx6-iCre mice on a swiss genetic background (generous gift from Professor Gordon J. Fishell) were crossed with either RCE-EGFP on a swiss genetic background (generous gift from Professor Gordon J. Fishell) or AI14-tomato (B6. Cg-Gt(ROSA)26Sortm14(CAG-tdTomato)Hze/J, The Jackson Laboratory, USA) on a swiss genetic background reporter mice to yield Lhx6-EGFP or Lhx6-tomato mice. For experiments, transgenic mice Lhx6-iCre$^{+/−}$; RCE-EGFP$^{+/−}$ were generated by crossing Lhx6-iCre$^{+/−}$;RCE-EGFP$^{−/−}$ with wt swiss mice (CE Janvier, France). For GAD immunocytochemistry, we used Gad1- GFP on a swiss genetic background knock-in mice where GFP replaces the first exon of the *Gad1* gene. For optogenetic experiments, hemizygous ChAT-ChR2-EYFP mice (6. Cg-Tg(Chat-COP4*H134R/EYFP,Slc18a3)6Gfng/J, The Jackson Laboratory) on a C57BL/6J genetic background were crossed with either C57BL/6J mice (The Jackson Laboratory) or Lhx6-tomato (first generation from hemizygous Lhx6-Cre mice crossed with homozygous AI14-tomato mice, Lhx6-iCre$^{+/−}$;AI14-tomato $^{+/−}$). Genotypes of the experimental mice were ChAT-ChR2-EYFP$^{+/−}$ or ChAT-ChR2-EYFP$^{+/−}$; Lxh6-iCre$^{+/−}$:AI14-tomato$^{+/−}$. We used both male and female mice. All experiments were performed in agreement with the European community council directives (2010/63/UE) and approved by the local ethics committee (D13 055 19).

**Chronic lesion of the dopaminergic innervation of the dorsal striatum.** The dopaminergic innervation of the left dorsal striatum of Lhx6-iCre;RCE-EGFP mice (25–40 g, aged postnatal day 30 (P30) to P40) was destroyed by local stereotaxic injection of 6-OHDA (Sigma-Aldrich, Inc., USA) under 5% ketamine (Imalgène® 1000, Merial SAS, France)/2.5% xylazine (Rompun® 2%, Bayer SAS, France) anesthesia (10 μl g$^{-1}$, intraperitoneally (i.p.)). Two microinjections of 6-OHDA were performed through a NanoFIL syringe (outside diameter, 135 μm; World Precision Instruments, USA) placed into the left dorsal. For sham-operated animals, we injected an equivalent volume of vehicle (sterile saline 0.9%), ascorbic acid 0.05%, pH 7.4. We performed in vitro recordings 15–20 days after lesion. The efficacy of the 6-OHDA-induced lesion of dopaminergic terminals in the striatum was determined 10 days before the recording session by apomorphine-induced rotation (0.5 mg kg$^{-1}$ in 0.1% ascorbic acid, i.p.)[44]. We checked the extent of the lesion after the recording session by immunohistochemical visualization of tyrosine hydroxylase (TH) in the striatum (see Supplementary Fig. 11).

**Slice preparation.** P30–P40 mice (25–40 g, aged P30–P40, both sexes) were anaesthetized and killed by decapitation. The brain was rapidly removed and placed in an oxygenated ice-cold saline buffer. Sagittal 300–380 μm-thick slices

**Fig. 4** Bumetanide restores GABAergic inhibition in CGINs. **a** Cell-attached recordings of current through single GABA$_A$ channels with GABA (1 μM) in patch pipette in CGIN at different holding potentials. **b** I–V relationships of the currents through single GABA$_A$ channels in CGINs in control and 6-OHDA-treated striatum; their reversal potential corresponds to DF$_{GABA}$. **c** Summary plot of DF$_{GABA}$ of CGINs inferred from single GABA$_A$ channels recordings. **d** Cell-attached recordings of single NMDA channel current with NMDA (10 μM) in patch pipette in CGINs at different holding potentials **e** I–V relationships of the currents through NMDA channels in CGINs in control and 6-OHDA-treated striatum; their reversal potentials correspond to resting membrane potentials (V$_R$). **f** Summary plot of V$_R$ of CGINs inferred from the reversal of single NMDA channels recorded in cell-attached mode. **g** Isoguvacine (10 μM) inhibited CGINs spontaneous activity (cell-attached recordings) in control, **h** but not in 6-OHDA-treated mice. **i** Bumetanide treatment restored inhibition in 6-OHDA-treated mice. **j** Effects of isoguvacine on spike frequency (normalized to control). **k** Pooled instantaneous frequency distributions of CGIN spontaneous spikes in control and 6-OHDA-treated mice with or without bumetanide treatment. **l** Corresponding mean frequencies of CGINs spiking activity. All means ± SEM. **c**, **f** Significance was determined by two-tailed, unpaired Student's t-test; see Supplementary Tables 10,11 for statistics. **j**, **l** Data sets were analyzed using one-way ANOVA followed by Fisher's least significant difference (LSD) post-hoc test; see Supplementary Tables 12,14 for statistics; **P < 0.01, ***P < 0.001. NS, not significant

were cut with a vibratome in ice-cold choline solution containing (in mM): 118 choline chloride, 2.5 KCl, 0.7 $CaCl_2$, 7 $MgCl_2$, 1.2 $NaH_2PO_4$, 26 $NaHCO_3$, and 8 glucose oxygenated with 95% $O_2$ and 5% $CO_2$. Before recordings, slices were incubated in artificial cerebrospinal fluid (ACSF) solution containing (in mM): 125 NaCl, 3.5 KCl, 0.5 $CaCl_2$, 3 $MgCl_2$, 1.25 $NaH_2PO_4$, 26 $NaHCO_3$, and 10 glucose, 300 mOsm, equilibrated at pH 7.3 with 95% $O_2$ and 5% $CO_2$ at room temperature

(RT) (22–25 °C) for at least 2 h to allow recovery. For the recordings, we used an ACSF of the same composition but containing 2 mM $CaCl_2$ and 1 mM $MgCl_2$.

**Patch-clamp recordings**. Slices were transferred to the recording chamber and perfused with oxygenated recording ACSF at 3 ml min$^{-1}$ at RT (22–25 °C).

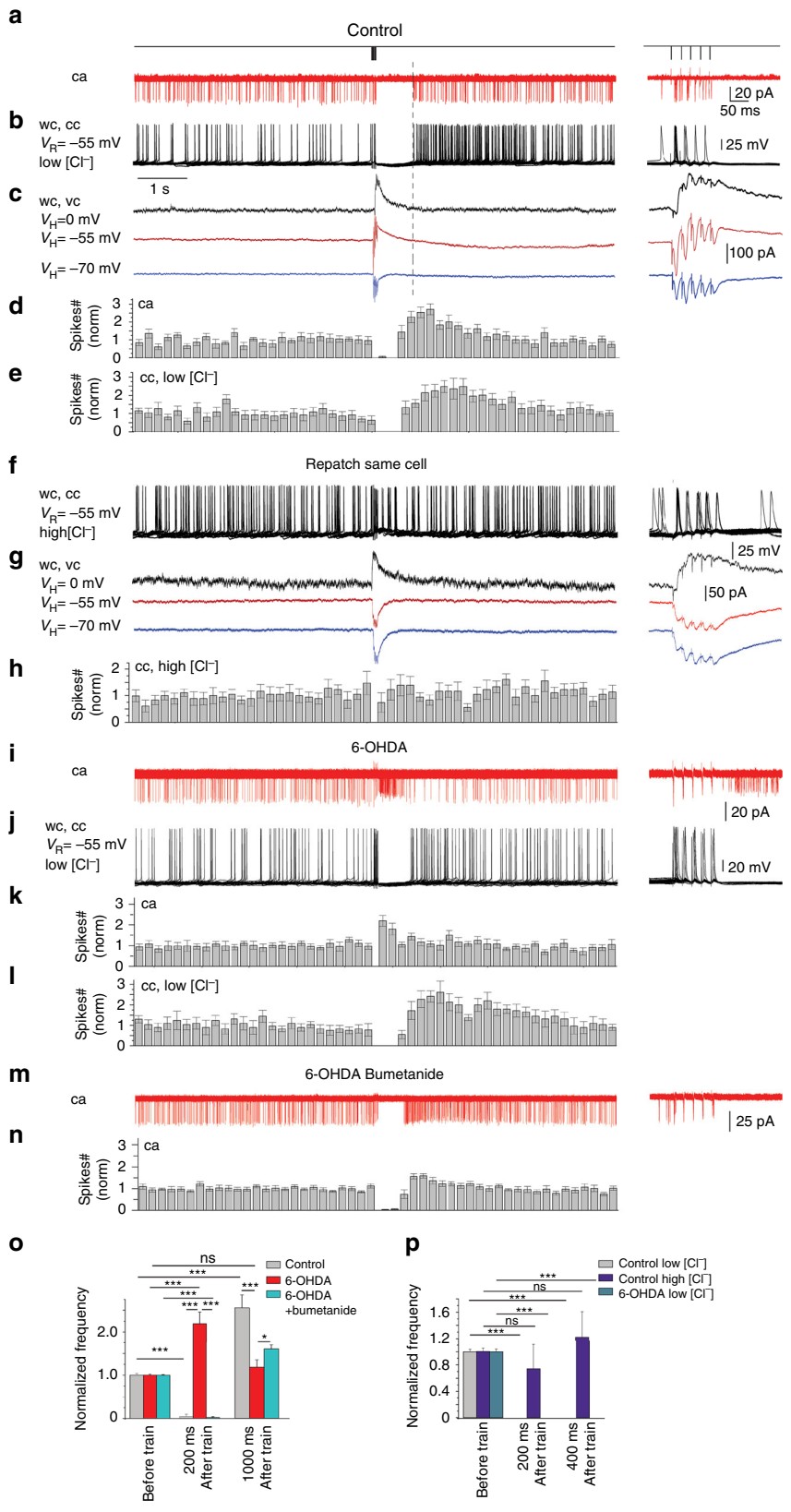

Neurons were visualized using infrared differential interference contrast microscopy. Patch pipettes had resistances of 5–7 MΩ when filled with the "low" chloride intracellular solution (in mM): 130 K-gluconate, 10 Na-gluconate, 7 NaCl, 4 MgATP, 4 phosphocreatine, 10 HEPES, and 0.3 GTP (pH 7.3 with KOH, 280 mOsm). "High" chloride intracellular solution (in mM) contained: 107 K-gluconate, 23 KCl, 10 Na-gluconate, 7 NaCl, 4 MgATP, 4 phosphocreatine, 10 HEPES, and 0.3 GTP (pH 7.3 with KOH, 280 mOsm, 5–7 MΩ). Equilibrium potentials of chloride ions were $-75$ mV for the "low" and $-38$ mV for the "high" chloride solutions (25 °C). Liquid junction potentials were calculated using Clampex's Junction Potential Calculator (16 mV in "low Cl" solution vs. 12.4 mV in "high Cl" solution, $\Delta 3.6$ mV) and corrected by the Pipette Offset circuitry of the amplifier. Giant GABAergic currents were recorded with a Cs-based solution (in mM): 120 Cs-gluconate, 13 CsCl, 1 CaCl$_2$, 10 HEPES, 10 EGTA (pH 7.3 with KOH, 290 mOsm, 5–7 MΩ).

Biocytin (final concentration of 0.3–0.5%) was added to pipette solutions to label recorded neurons. Cells with leakage current more then 40–50 pA were discarded.

To determine cell excitability, we recorded voltage responses (cc mode) to 1 s current steps of $-200$ to $+200$ pA (10 pA increment, 3 s interval between each step). Current–voltage (I–V) relationships were established to calculate input resistance of cells. For the hyperpolarization-activated sag, we measured the ratio between the peak and minimal amplitudes of the voltage response. Rheobase current, the minimal current that elicits an AP, was determined using a 10 pA step. AP duration (half width) was measured at half of the maximal amplitude. After-hyperpolarization was measured at the peak.

Patch pipette solution for recordings of single GABA$_A$ channels contained (in mM): NaCl 20, KCl 5, TEA-Cl 20, 4-aminopyridine 5, CaCl$_2$ 0.1, MgCl$_2$ 10, glucose 10, Hepes-NaOH 10 buffered to pH 7.2–7.3., and for recordings of single N-methyl-D-aspartic acid (NMDA) channels: nominally Mg$^{2+}$-free ACSF with NMDA (10 μM) and glycine (10 μM). Pipettes had a resistance in the range of 7–10 MΩ. We performed conventional ca recordings under visual control. With 1 μM GABA in the pipette solution, after gigaseal formation ($>3$ GΩ) currents through GABA$_A$ channels of 1 pA were immediately visible at the potential $+50$ mV. Currents through GABA$_A$ channels were recorded from $-50$ to $+50$ mV with 10 mV increments for 1–2 minutes for each holding potential, depending on the ongoing frequency of GABA$_A$ channels openings, to obtain at least 20 single-channel openings for each potential. After the recording was completed, we repatched the same cells with the NMDA-containing solution. We performed ca recordings of NMDA channels activity at various pipette potentials from $-80$ to $+20$ mV with 10 mV increments for 1–2 min. We performed analysis of single channel currents and I–V relationships using Clampfit 10.6 (Axon Instruments, Union City, CA).

**Identification of recorded striatal neurons**. We identified CINs by their large soma ($\geq 30$ μm diameter), thick primary dendrites, and intrinsic membrane properties. Hyperpolarizing pulses produced an initial hyperpolarization followed by a depolarizing sag followed by rebound spikes after pulse cessation. Post-hoc examination of biocytin-filled neurons confirmed the cholinergic identity of recorded cells. FS interneurons discharged AP in high-frequency trains. In addition to fast spikes, PLTS interneurons displayed low-threshold spikes when depolarized from potentials near $-70$ mV or after cessation of hyperpolarizing pulses (see Fig. 1b).

**Extracellular stimulation**. To stimulate the cortico-striatal pathway, a bipolar Ni–Cr electrode was positioned on the surface of the Corpus callosum just below M1 motor cortex (see Supplementary Fig. 4). Single or trains (5 stimuli, 50 Hz) of current pulses (25–50 μA, 100 μs)) were delivered every 200 s through the constant current bipolar stimulus isolator A365 (World Precision Instruments).

**Optogenetic stimulation**. Photo-stimulation of the light-sensitive cation channel, ChR2[45,46], was performed using a micro-mirror array system (Digital Micromirror Device, or DMD), which allows arbitrary spatial patterns of illumination[47] and thereby photo-stimulation of spatially determined populations of presynaptic neurons[48–50]. Photo-stimulation was performed with a DMD (Mosaic, Andor Technology, UK), using a high-power light-emitting diode (CoolLED pE-4000,

CoolLED Ltd., UK). PatchMaster and Andor iQ3 softwares were used to synchronize the light stimulation and electrophysiological recordings. A 490 nm LED was used to activate ChR2 and thereby photo-stimulate CINs. Spatially patterned illumination was generated by the DMD and projected onto the slice via the microscope objective. The spatial pattern of illumination was digitally controlled with microsecond time resolution, by toggling each of the 777,600 (1080 × 720) micro-mirrors within the array. A brief light pulse (2–10 ms) reliably evoked APs in CINs (Supplementary Fig. 8).

**Immunohistochemistry and biocytin-filled cells revelation**. To visualize dopaminergic innervation of the striatum, we performed immunohistochemistry of TH of slices in which we performed recordings and those immediately medial or lateral. After overnight fixation at 4 °C in Antigenfix (Diapath, Italy), sections were washed three times in phosphate-buffered saline (PBS, Life Technologies, USA) and incubated for 30 min at RT in PBS containing 5% normal goat serum (NGS, Jackson ImmunoResearch, Inc., USA) and 0.3% Triton X-100 (Sigma-Aldrich). Then they were incubated overnight at 4 °C with a Rabbit anti-TH polyclonal antibody (1:1000, Pel-Freez Biologicals, USA, ref. P40101-150) diluted in the same buffer. After wash with PBS, they were incubated at RT for 2 h with a Goat anti-rabbit antibody coupled to Alexa Fluor 647 (1:500, Invitrogen, USA) diluted in PBS containing 5% NGS. They were washed three times 10 min in PBS and cover-slipped using Fluoromount-G$^{TM}$ (Electron Microscopy Science, USA) as mounting medium. Electrophysiological data were taken into account only when a severe loss of TH immunoreactivity was present in the dorsal striatum in which recordings were performed (Supplementary Fig. 11).

To identify cholinergic cells, we performed immunohistochemistry using a Goat anti-ChAT antibody (1:1000, ref. AB 114 P, Millipore, USA). After 1 h incubation at RT in PBS containing 5% normal donkey serum (NDS) and 0.3% Triton X-100, slices were incubated overnight at 4 °C in the same buffer containing the primary antibody. After washing thrice for 10 min in PBS, slices were incubated for 2 h at RT with a Donkey anti-Goat antibody coupled to CY3 (1:500, Millipore) in PBS containing 5% NDS. The same procedure was followed to perform Lhx6 staining using a Mouse anti-Lhx6 (1:50,000, Santa Cruz Biotechnology, USA, ref. sc-271433) and a Donkey anti-mouse antibody coupled to Alexa Fluor 647 (1:500, Invitrogen). To determine the GABAergic phenotype of ChAT cells, we performed a co-staining between VGAT, GAD65-67, and ChAT on striatal slices from wt and Lhx6-iCre;RCE-EGFP mice by using a Guinea Pig anti-VGAT antibody (1:500, ref 1310004, Synaptic System, Germany) and a Rabbit anti-GAD65-67 (1:500, ref AB1511, Millipore), respectively. After 30 min incubation at RT in PBS containing 1% bovine serum albumin (BSA) and 1% NDS and 0.3% Triton X-100, slices were incubated overnight at 4 °C in PBS, 1% BSA, and 0.1% Triton X-100. Slices were finally incubated for 1 h at RT with a Donkey anti-Goat Alexa Fluor 568 (1:500, The Jackson Laboratory) and a Donkey anti-guinea pig Alexa Fluor 647 (1:500, The Jackson Laboratory) or a Donkey anti-rabbit Alexa Fluor 647 (1:500, The Jackson Laboratory). To verify the selective expression of EYFP in ChAT-positive cells in ChAT-ChR2-EYFP and ChAT-ChR2-EYFPxLhx6-iCre;AI14-tomato mice, we used the same protocol as above. Striatal coronal slices were incubated with ChAT primary antibody coupled with a Donkey anti-Goat Alexa Fluor 633 (1:500, The Jackson Laboratory).

Quantification of ChAT-, VGAT-, GAD65-67-, EGFP-, and EYFP-positive cell number and assessment of colocalization was performed with Image J Software throughout the entire z-dimension of six or eight sections per brain.

To reveal biocytin, slices were fixed overnight in Antigenfix at 4 °C, slices were washed three times in PBS, and after a 1 h pre-incubation in PBS containing 5% NGS and 0.3% Triton X-100, they were incubated overnight at 4 °C with streptavidin coupled to Alexa Fluor 555 (1:500, Life Technologies) in the same buffer. Slices were then washed three times 10 min in PBS and cover-slipped using Fluoromount-G$^{TM}$ as mounting medium. To verify whether the recorded cells were located in the 6-OHDA-lesioned region of the striatum, we combined the two types of post-hoc immunohistochemistry.

For biocytin-filled cells and TH staining detection, mosaics were acquired using an Axio Imager Z2 microscope (Carl Zeiss GmbH, Germany) using the × 10 objective (numerical aperture (NA) 0.3), and the 533–558 nm (beam splitter 570) and 625–655 nm (beam splitter 660) wavelength for excitation of Alexa Fluor 555 and 647 (spectral detection 570–640 nm and 665–715 nm, respectively).

**Fig. 5** Polarity of GABA action determines the pause-rebound response evoked in CGINs by cortical stimulation. **a–n** Left: Stimulation protocol, representative 12–18 superimposed consecutive CGINs responses to cortical stimulation and mean frequency histograms; right: responses within the train at extended time scale. Abbreviations: ca, cell-attached; wc, whole-cell; cc, current; vc, voltage clamp modes at different holding ($V_H$) or resting ($V_R$) membrane potential with "low" or "high" [Cl$^-$]$_i$ containing pipettes. **a–h** Control mice. Pause-rebound response recorded in **a** ca and in wc in the same CGIN with "low" chloride containing pipette solution in **b** cc or **c** vc modes. Mean frequency histograms for **d** ca and **e** cc recordings. **f–h** Still same CGIN repatched with "high" [Cl$^-$]$_i$ solution Note that the pause response was abolished. **i–n** 6-OHDA-treated mice: same experiments as in **a, b, d, e. i, k** Note the absence of the pause response in ca, which reappeared in wc–cc with "low"[Cl$^-$]$_i$ **j, l. m, n** restoration by bumetanide of the pause-response in ca. **o, p** Mean number of spikes during time windows after train stimulation (0–200 ms, 200–400 ms, 800–1000 ms) normalized to spikes counts before train, and recorded in **o** ca or **p** wc–cc. All means ± SEM. **o, p** Data sets were analyzed using one-way ANOVA followed by Fisher's least significant difference (LSD) post hoc test. See Supplementary Tables 16, 17 for statistics; *$P < 0.05$, ***$P < 0.001$. NS, not significant

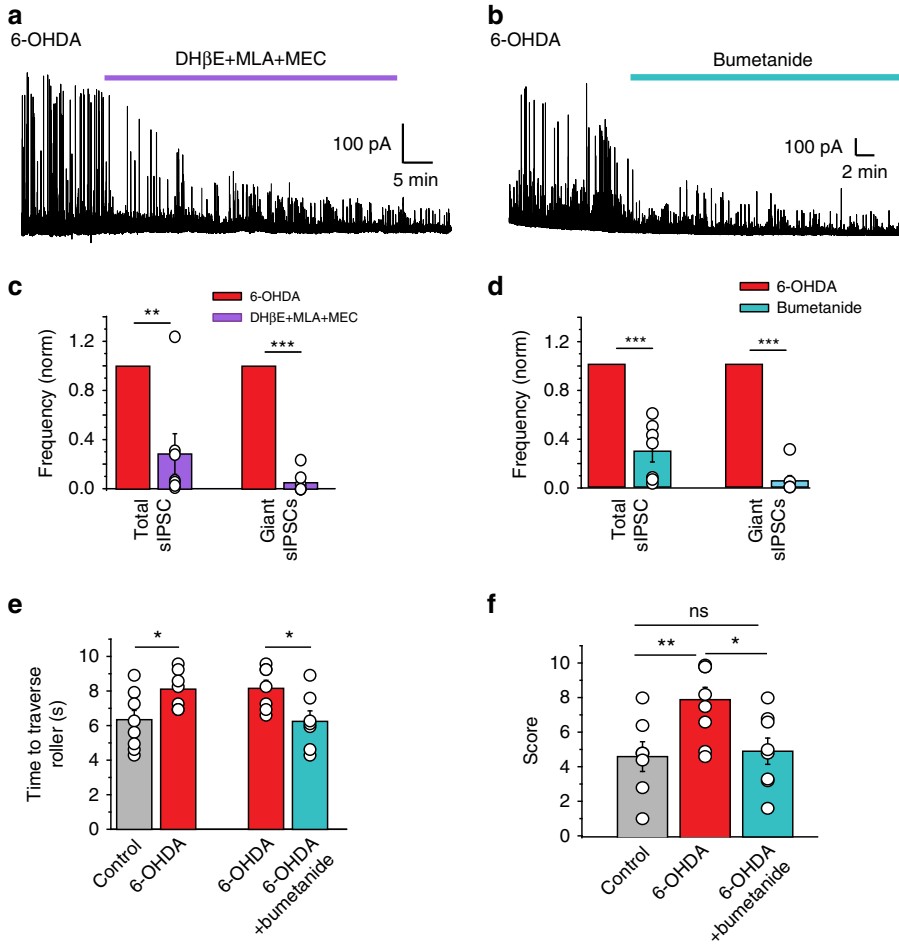

**Fig. 6** Bumetanide attenuates electrical network and behavioral effects of dopamine deprivation. Nicotinic receptor antagonists **a**, **c** or bumetanide **b**, **d** treatment blocked giant GABAergic PSCs recorded in SPNs (6-OHDA-treated mice) at $V_H = +10$ mV. **e**, **f** Behavioral tests. **e** Time to traverse roller in control and 6-OHDA-treated mice (same 6-OHDA-treated mice before and after bumetanide treatment). **f** Pole test score of the three different mouse groups. See Supplementary Movies 2-7. All means ± SEM. **c**, **d** Significance was determined by two-tailed, paired Student's t-test; see Supplementary Tables 19 and 20 for statistics. **e** Significance was determined by two-tailed unpaired (left set) or paired (right set) Student's t-test; see Supplementary Table 21 for statistics. **f** Data sets were analyzed using one-way ANOVA followed by Fisher's least significant difference (LSD) post-hoc test; see Supplementary Table 22 for statistics; *$P < 0.05$, **$P < 0.01$, ***$P < 0.001$. NS, not significant

For triple-labeled EGFP+-ChAT-Lhx6 cells detection, images (pixel size 0.321 μm) were acquired on a Zeiss LSM 800 confocal laser-scanning microscope (Carl Zeiss) using the 488, 555, and 647 nm lasers for excitation of EGFP, CY3, and Alexa Fluor 647 (spectral detection, 400–550 nm, 550-630 nm, and 630-700 nm, respectively). We used the ×40 objective (NA 1.4) and sequentially scanned each channel with a frame average 2. For ChAT, VGAT, and GAD65-67 immunostaining, images were acquired on a confocal laser scanning microscope Leica TCS SP5X equipped with a white light laser, a 405 nm diode for ultraviolet excitation, and 2 HyD detectors, using the ×63 objective, and the 568 nm and 647 nm wavelength for ChAT and VGAT or GAD65-67, respectively.

For biocytin-filled cell reconstructions, confocal images (pixel size 0.240 μm) were acquired on a SP5X Leica microscope (Leica Microsystems) using the 555 nm band of a white laser for excitation of Alexa Fluor 555 (spectral detection, 565–675 nm). Images were acquired at 400 Hz using a ×63 oil-immersion objective, pinhole set to "Airy 1", by scanning with a z step of 0.50 μm.

**Reconstruction of biocytin-filled neurons and Sholl analysis**. Only neurons with a complete dendritic arbor were analyzed. Stacks were imported in the open-source platform Fiji (https://fiji.sc/) and stitched together (https://imagej.net/Image_Stitching)[51]. Without any additional pre-processing, dendrite and axon reconstructions, as well as morphological metric measurements were performed using the semi-automated "Simple Neurite Tracer" plugin (http://imagej.net/Simple_Neurite_Tracer)[52]. Based on dendrites reconstructions, a Sholl analysis was performed using "Sholl Analysis" plugin (http://imagej.net/Sholl_Analysis)[53].

**Single-cell RT-qPCR**. Single-cell RT-qPCR was performed according to modified protocols described in ref.[54]. Sterile patch pipettes had a tip outer diameter of ~

2–3 μm and a resistance of 0.7–1.5 MΩ when filled (3–5 μL) with the "low" chloride intracellular solution prepared with RNAse-free water (Roche, USA) and containing 1000 units per mL of Protector RNase Inhibitor (Roche). Cytoplasm and nucleus of identified EGFP+ cells were collected into the patch pipette under visual control. Appropriate controls for possible DNA and RNA contamination were performed when harvesting cells, Reverse Transcription and qPCR reaction: (i) RT-q PCR was performed in individual cells (not pulled data set, as typical practice for these studies) to avoid false positive result due to contamination; (ii) SPNs were used as negative control; (iii) water was tested instead of cell DNA in the RT-qPCR mix to check contamination in the RT-qPCR mix; (iv) samples with no reverse transcriptase to check DNA contamination in the reverse transcription mix; (v) pipette immersed in slice without patching and then same procedure to check contamination in the pipette solution and the medium of recording.

The content of the patch pipette was expelled into 11.4 μL of re-suspension buffer containing 20 units of Protector RNase Inhibitor and 60 μM random hexamer primer (Roche). After 10 min of denaturation at 65 °C, reverse transcription was performed using Transcriptor high-fidelity cDNA synthesis kit (Roche) on a SimpliAmp Thermal Cycler (Applied Biosystems, USA) following the kit procedures and then kept on ice until the PCR analysis. The PCR analysis was conducted using the LightCycler 480 Real-Time PCR System (Roche) in a total volume of 15 μL containing 3.75 μL of template DNA, 7.5 μL LightCycler 480 Probes Master (Roche), and 1.5 μL of selective Taqman® gene expression assay for ChAT or GAD65, or Lhx6 or Lhx7 mRNAs with hypoxanthine phosphoribosyltransferase (HPRT) mRNA (Life Technologies). HPRT was used as housekeeping or reference gene (see Supplementary Table 23 for specific references), because its expression level is close to that of the target genes in the cell of interest. The reactions were subjected to an initial incubation at 95 °C for 10 min, followed by 50 cycles of 95 °C for 10 s, annealing and extension for 40 s at 60 °C.

Fluorescein amidite (FAM) filter detection was 483 nm for excitation and 533 nm for emission, and 2′-chloro-7′phenyl-1,4-dichloro-6-carboxy-fluorescein (VIC) filter detection was 523 nm for excitation and 568 nm for emission. Dual-color relative quantification analysis was performed using LightCycler®480 Instrument Software version 1.2 (Roche, Germany). The heat map shows threshold cycle (Ct) of the target gene normalized to the Ct of the reference gene (HPRT) encoded by color (white to red).

**Mice behavioral tests**. For evaluating mice bradykinesia and motor coordination, we used the pole test. Mice descend a vertical wooden pole (50 cm long and 1 cm diameter) leading to their home cage[55]. We performed this test 8–9 weeks following 6-OHDA lesion when animals were at least 3 months old. For the treatment of randomly chosen group, Bumetanide (2–2.5 mg kg$^{-1}$) was given in drinking water for 5 weeks before the test.

First day of training: To train mice to walk down to their home cage from the top of the pole, mice were first placed head-down on the top of the pole (three trials). Then they performed a minimum of five trials with head up to train them to turn 180° at the top of the pole and to descend the pole.

Second day of training: 24 h after the first training, mice performed five trials with their head-up on the top of the pole.

Test day: 24 h after second training, we videotaped the tests and measured the total duration of the descent and scored it, as previously described[56] with modifications. The scores were as follows: 1 for a descent lasting 1–3 s, 2 for 4–6 s, 3 for 7–9 s, 4 for 10–12 s, 5 for 13–15 s, 6 for 16–18 s, 7 for 19–21 s, and 8 for 22– 24 s. A score of 9 was given when the mouse descended part way and fell the rest of the way, 10 if it slid down the pol, and 11 if it fell from the top of the pole.

Five trials were performed for each mouse and the mean data across the trials calculated. The videos were viewed and scored by an investigator who was blind to the animal condition.

To assess motor coordination of mice we used an apparatus set up by Dr. Paikan Marcaggi. This apparatus first consisted of a horizontal flat beam (114.5 cm long, 2.5 cm wide), held 55 cm above the bench surface by a column at each end. After a few trials on the beam, we added the roller bar in the middle of the beam as explained below. In order to cushion the eventual fall of mice during testing, a soft padded surface was placed at the base of the apparatus. After 2 days of training, the test was performed as previously described for the challenging beam test[57] with modifications.

First roller test was carried out 8–9 weeks following 6-OHDA lesion when animals were at least 3 months old. To do pre- and post-Bumetanide treatment measurements, after the first roller test, a randomly chosen group of mice received the Bumetanide treatment (2–2.5 mg kg$^{-1}$) in drinking water for 5 weeks, then all groups underwent second (post treatment) roller test.

First day of training: Mice performed three assisted trials to pass the length of the beam and arrive at their home cage located at one end (there was no roller bar on the beam at this step). When the mouse was placed on the apparatus for the first time, we let it move around and sniff to become oriented to the apparatus. If the mouse turned around, we gently redirected it to the intended direction, but if it continued turning around, we brought the home cage close to the mouse. When the mouse tried to get into the home cage, we moved it back so the mouse could not enter, but had to do a step forward. We continued to do this all the way down the beam. At the end of the beam, we let the mouse enter the home cage. This was the first assisted trial. All mice performed a minimum of three trials to be sure that they can walk the length of the beam on their own. Afterwards, we placed a 36.5 cm roller bar in the middle of the beam to challenge the mice, and they performed a minimum of three more trials getting assistance with climbing up the roller and traversing it to be sure that the mouse was able to perform the task on its own.

Second day of training: 24 h after the first training, mice performed three trials on the roller bar.

Test day: Finally, 24 h after the second training, we videotaped the mice while traversing the roller during three consecutive trials. The videos were viewed and scored for the time to traverse the roller bar, by an investigator who was blind to the animal condition. Mean data across the three trials was calculated.

**iDISCO protocol**. Samples were dehydrated in a graded series (20%, 40%, 60%, 80%, and 100%) of methanol (Sigma-Aldrich) diluted in PBS, during 1.5 h each at RT. They were then incubated overnight at RT on a platform shaker in a solution of PBSG-T (PBS containing 0.2% gelatin (Sigma-Aldrich), 0.5% Triton X-100, and 0.02% Sodium-Azide (Sigma-Aldrich)) for 2 days. Next, samples were transferred to PBSG-T containing the primary antibodies (Chicken IgY anti-GFP, 1:4000, AVES USA, AB144 Goat anti-ChAT, 1:500, Merck-Millipore, USA) and placed at 37 °C, with rotation at 100 r.p.m., for 15 days. This was followed by six washes of 1 h in PBSG-T at RT. Next, samples were incubated in secondary antibodies (Donkey anti-chicken Alexa Fluor 647 and Donkey anti-Goat Alexa Fluor 555, 1:500, Jackson ImmunoResearch) diluted in PBSG-T overnight for 2 days at 37 °C. After six washes of 1 h in PBSG-T at RT, samples were stored at 4 °C in PBS until clearing.

Tissue clearing was performed according to the clearing procedure reported earlier[58,59] with some modifications. Briefly, all incubation steps were performed at RT using a 15 ml centrifuge tube (TPP, Dutscher, France) covered with aluminum foil to avoid exposure to light. Samples were dehydrated in a graded series (20%, 40%, 60%, 80%, and 100%) of methanol diluted in PBS, during 1 h. This was

followed by a delipidation step of 20 min in dichloromethane (DCM; Sigma-Aldrich). Samples were then transferred to 100% DCM until they have sunk. Finally, samples were cleared overnight in dibenzylether (DBE; Sigma-Aldrich) and stored in polypropylene tubes filled with DBE, at RT in the dark.

A 3D imaging was performed with an ultramicroscope (LaVision BioTec GmbH, Germany) using ImspectorPro software (LaVision BioTec). The light sheet was generated by a laser (wavelengths 488, 555, and 647 nm, Coherent Sapphire Laser, LaVision BioTec), and two cylindrical lenses. A binocular stereomicroscope (MXV10, Olympus, Japan) with an objective × 2 (MVPLAPO, Olympus) was used at magnification × 2.5. Samples were placed in an imaging tank made of 100% quartz (LaVision BioTec) filled with DBE and illuminated from the side by the laser light. Images were acquired with a PCO Edge SCMOS CCD camera (2560 × 2160 pixel size, LaVision BioTec). The step size between each image was fixed at 2 µm. Three-dimensional image quantifications and movies were generated using Imaris x64 software (version 8.4.1, Bitplane, Switzerland). Stack images were first converted to Imaris file using ImarisFileConverter. Each resulting Imaris file was 16-bit images. A 3D reconstruction of the sample was performed using "volume rendering" (Imaris). Striatal segmentation was based on Allen Mouse Brain Atlas (Allen Institute[60]) and performed on autofluorescence channel (488). Briefly, "surface" tool were used to delimitate the structure and create a mask of left and right striatum. A second segmentation was processed to isolate the dorsolateral part of the structure. Three-dimensional pictures and movies were generated using the "snapshot" and "animation" tools.

The left segmented striatum obtained with Imaris was exported in TIF image sequence and a background substraction (rolling ball radius: 5.0 pixels) processed for each channels, using Fiji Software[61]. To reduce the risk of counting the same ChAT$^+$ cell several times in the Z-Stack, a substack was generated containing substituted slices by black slices every 20 µm. Cells were automatically counted in each slice with Cell Profiler cell image analysis software (Broad Institute Cambridge, USA). For each channel of TIF images, mean image intensity was measured using "MeasureImageIntensity" module and cells were identified with "IdentifyPrimaryObjects" module. The Min and Max typical diameter of objects was set between 8 and 30 pixel units for ChAT$^+$ cells and between 3 and 25 pixel units for EGFP$^+$ cells. Threshold was corrected by the previously measured mean intensity. Resulting detected objects were filtered based on their eccentricity (range between 0.2 and 0.8) using "FilterObjects" module. Then, using mask images generated with the aforementioned module, colocalization between ChAT and EGFP channels was evaluated with "RelateObjects" and "MaskObjects," in order to determine the number of overlapping objects. Finally, data were exported to a spreadsheet containing the number of quantified cells for each staining and co-localization.

**Drugs**. For in vitro experiments, SR 95531 hydrobromide (gabazine 5 µM, Tocris Bioscience, UK, Ref. 1262), NMDA (10 µM, Tocris Bioscience, UK, Ref. 0114), 2,3-dihydroxy-6-nitro-7-sulfamoyl-benzo[f]quinoxaline-2,3-dione (10 µM, NIH generous gift), GABA (10 µM, Sigma-Aldrich), isoguvacine (10 µM, Sigma-Aldrich, Ref. G002), DL -2-Amino-5-phosphonovaleric acid (40 µM, Sigma-Aldrich, Ref. A5282), and bumetanide (10 µM, Sigma-Aldrich, Ref. B3023) were directly added to the perfusion solutions. For ca experiments, slices were treated with bumetanide for 40 min before and during recordings. Cocktail of nicotinic receptor antagonists included mecamylamine hydrochloride (10 µM, Tocris Bioscience, Ref. 2843/10), methyllycaconitine citrate (0.1 µM, Tocris Bioscience, Ref. 1029/5), and dihydro-β-erythroidine hydrobromide (10 µM, Tocris Bioscience, Ref. 2349/10). For in vivo experiments, bumetanide pretreatment (3 mg kg$^{-1}$) was given to mice in drinking water during 5 weeks.

**Statistical analysis**. Data from the electrophysiological and behavior studies were analyzed with two-tailed $t$-test or one-way analysis of variance (ANOVA)–Fisher's least significant difference (LSD) post-hoc tests. For morphological metrics, we used one-way ANOVA–Kruskal–Wallis tests followed by Dunn's multiple comparison post-hoc tests. For Sholl analysis of dendrites, Kolmogorov–Smirnov tests were performed. Analyses were performed with Prism 6 (GraphPad Software Inc., USA) or OriginPro (OriginLab, USA). All data are presented as means ± SEM. *$P$ < 0.05; **$P$ < 0.01; ***$P$ < 0.001.

**Data availability**. All relevant data are available from the corresponding authors upon reasonable request.

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

## Acknowledgements

We are grateful to Dr. P. Marcaggi for the apparatus to assess motor coordination, to Dr. A. Chedotal and M. Bee for help with the iDISCO clarification technique, and to Professor G. Fishell for transgenic mice. We are also grateful to Drs F. Michel, D. Ferrari, and C. Cardoso for suggestions, Dr. G. Chazal for help with immunohistochemistry, Dr. S. Platel for help in the RT-qPCR technique, and to R. Ahamada and T.-T. Bui for technical support. Financial support is acknowledged from Amidex, Neurochlore, and B&A Therapeutics.

## Author contributions

Y.B.-A. and C.H.: project leaders. Y.B.-A., C.H., N.L. and N.B.: design of experiments. Y. B.-A., C.H., N.L. and N.B.: writing paper. N.L. and L.A.G.-C.: electrophysiological experiments. S.E.: behavioral studies. B.R. and R.C.: iDISCO. M.B.-G. and A.P.-B.: PCR experiments. A.D. and N.O.: morphology. R.C., B.R. and N.O.: immunohistochemistry. N.O.: animal's surgery. N.L., S.E., R.C., L.A.G.-C., B.R., M.B.-G., A.D. and A.P.-B.: analysis and statistics. All authors discussed the results and commented on the manuscript.

## Additional information

**Competing interests:** N.L., S.E., R.C., L.A.G.-C., A.D., B.R., M.B.-G., N.O., and Y.B.-A. are salaried of Neurochlore or B&A Therapeutics. N.L., N.B., Y.B.-A., and C.H. are shareholders of Neurochlore. Y.B.-A. is CEO of Neurochlore and B&A Therapeutics. A.P.-B. declares no competing interests.

