## [Peer Review File · Nature Communications]

Reviewers' comments:

Reviewer #1 (Remarks to the Author):

The manuscript by Lozovaya et al. described a novel finding that many striatal cholinergic interneurons co-expressed GAD, and were capable of releasing both ACh and GABA. In addition, such co-transmission is impaired in a mouse model of Parkinson's disease (6-OHDA injection model). Mechanistically, the Cl transporter function is involved. The finding is novel and very interesting. The data are presented with high quality, and convincing. Therefore, we recommend publication in the journal.

However, we do have a few concerns and suggestions:

1. The previous work using ChAT-ires-Cre mouse and optogenetic tools only revealed di-synaptic inhibition, most likely by recruiting putative GABAergic neurons. Here, the authors used ChAT-ChR2-YFP transgenic mouse, and *Ihx6*-iCre mouse. To better integrate with existing literature, I would recommend the authors to perform characterization using similar approach shown in Fig 2., but instead, using ChAT-ires-Cre;Ai14 mouse combined with staining for GABAergic neuron markers. This would solve the discrepancy. Potentially ChAT-ChR2 mouse and ChAT-ires-Cre mouse may label different subsets of cholinergic interneurons.

2. Previous studies from Paul Bolam, and Jim Surmeier groups have shown that CINs receive large amount of thalamic inputs. Here, the authors used cortical stimulation. It is worthy discussing whether two different cholinergic interneurons have different afferent inputs (cortical vs thalamic).

3. When using low Cl and high Cl internals, the reported values in mV may be complicated by their different junction potentials. How much were the junction potential when using low and high Cl internals? Was junction potential corrected in the reported values?

Reviewer #2 (Remarks to the Author):

The manuscript by Lozovaya et al. advances two very interesting ideas. First, they argue from lineage markers that there are two populations of striatal cholinergic interneuron: one that releases acetylcholine alone and another that co-releases GABA. In addition, they differ in dendritic anatomy and possibly physiology, although that is not as clear. Second, the authors argue that following lesioning of the dopaminergic innervation of the striatum, the intracellular chloride concentration rises specifically in interneurons that co-release GABA, causing GABAergic signaling to become less inhibitory, causing aberrant activity patterns and promoting parkinsonian behavior. Although the results are intriguing, only the first half of the paper is convincing. Indeed, there appear to be two populations of cholinergic interneuron. That said, the authors need to provide a better characterization of these two particularly with regard to their physiology and ability to co-release GABA. The data on this latter point is particularly problematic. Insofar as the second idea is concerned, the results

are far from convincing that the chloride reversal potential changes in a subpopulation of interneurons and that this change is causally linked to the parkinsonian motor phenotype. These and other concerns are outlined below.

Major concerns:

- Based upon the scRT-PCR data in Figure 1, there is no relationship between GAD65 expression and Lhx6 expression. This is not clear from reading the text. Wasn't this the starting hypothesis?
- Was vGAT mRNA present? Previous work in other cell types has reported co-expression of ChAT, GAD and vGAT expression as I recall. What about NKCC1 mRNA?
- Cholinergic interneurons have previously been shown to co-release glutamate and acetylcholine. Was vGluT3 co-expressed in either population?
- The evidence that cholinergic interneurons receive monosynaptic GABAergic input from other interneurons in the Lhx6+ class is not very convincing. Was the GABAergic response in interneurons evoked by optical stimulation of Lhx6+ interneurons sensitive to ionotropic glutamate receptor antagonists? A more direct and simpler test is to bath apply TTX (to block conducted activity) and micromolar 4-AP and then optogenetically stimulate again. If there are interneuron synapses on the cell being recorded from, you'll see the synaptic response.
- In previously published work, the spiking rate of cholinergic interneurons in ex vivo slices from 6-OHDA lesioned mice is not significantly different from that found in unlesioned mice.
- An increase in dendritic surface area is not strong evidence for an increase in collateral connectivity. More direct evidence is necessary. Also, was the change in dendritic surface area with lesioning peculiar to Lhx6+ interneurons?
- The striatal network is highly interconnected with many GABAergic neurons that will be sensitive to bumetanide and isoguvacine. This makes the derivative outcome measure of spontaneous spiking rate of questionable value to the conclusions about the Cl⁻ reversal potential. A direct assessment of this value in Lhx6+/- interneurons using perforated patch techniques must be made. This measure should be made in control and lesioned mice in both interneuron subtypes.
- The cholinergic interneuron burst-pause pattern of responding to excitatory synaptic input has previously been shown to have several determinants. At higher burst frequencies and stronger stimulation intensities, GABAergic signaling is clearly one of these. So, the dependence of the pause upon the intracellular Cl⁻ concentration seen here is expected. It is important to show that the pause evoked with this protocol is dependent upon GABAA receptors. Is there any DA dependence to the pattern? Can the pause be self-induced by recurrent collaterals?
- Is there a change in NKCC1 expression in Lhx6+ interneurons following 6-OHDA lesioning?
- The work with the IPSPs in SPNs is a bit confusing. Previous work by the authors implicated PLTSIs – not cholinergic interneurons – as the source of these IPSPs in SPNs following 6-OHDA lesioning. In fact, didn't this previous work claim they were insensitive to nAChR antagonists?
- The behavioral experiments need to be more completely described. They cannot be reproduced from what is given. Also, the timing of drug administration needs to be clearly stated. The videos help but also raise concerns about the nature of the motor disability that

is alleviated by bumetanide.

- Aside from the ambiguity in the methods, the behavioral effects of bumetanide in the 6-OHDA lesioned mouse are impressive. However, it is not reasonable to attribute those changes to high intracellular Cl⁻ concentration in Lhx6⁺ interneurons. Bumetanide will increase GABAergic inhibition of many neurons in the striatum (and the brain). With the availability of cell type specific manipulations using optogenetics or chemogenetics, there isn't any excuse for not keeping the evidentiary bar high. In fact, the chemogenetic approaches would even allow selective elevation in chloride conductance.
- Lastly, the statistical approaches used for display and significance testing

Minor concerns:

- The electrical stimulation protocol does not ensure that thalamostriatal glutamatergic axons are not being stimulated. Not only is current spread an issue, but most thalamostriatal neurons have a cortical axon collateral. The authors should simply refer to the stimulation as capable of activating glutamatergic axons.
- At what temperature were there recordings made?
- How were PLTSIs identified?
- In figure 3, were AMPA and NMDA receptor blockers present? Doesn't the similarity in the delays of the GABA_A IPSP with optogenetic and electrical stimulation imply both are polysynaptic?
- Fig. 1: Is the apparent absence of a rebound in Lhx6⁻ interneurons simply a reflection of the weak pause? Is this rebound intrinsically generated or synaptic? Is the evoked increase in spiking different? In the raw data (j) it appears so.
- What controls were run for contamination? None are obvious in the records presented. mRNA for D1 dopamine receptors or substance P or enkephalin should be undetectable.
- Figure 5p is not justified.
- In figure 4c, why was time window different from figure a and b?

Reviewer #3 (Remarks to the Author):

The main findings of the study are: (i) that, striatal cholinergic interneurons (CINs) can be subdivided into 2 distinct subtypes that differ in terms of their morphology, intrinsic electrophysiological properties and postsynaptic responses (including a pause response) to activation of cortical inputs, (ii) that, one of these types, representing about 50% of all CINs also express the LIM homeobox protein Lhx6 and the GABAergic marker GAD65, (iii) the latter class of cells, thus termed GABAergic-CINs (GCINs), are GABAergic and cholinergic in terms of signaling properties (iv), 6-OHDA lesion of the nigrostriatal dopaminergic pathway increases the connectivity (GABAergic and cholinergic) of GCINs, dramatically changes the morphology of these cells and blocks the expression of the cortically evoked pause response due to abnormal intracellular chloride levels ([Cl]_i), (v) the application of a NKCC1 chloride transporter antagonist (bumetanide) restores normal [Cl]_i and the cortically evoked pause in GCINs, and finally, (vi) that, the same drug ameliorates symptoms of Parkinsonism in 6-OHDA lesioned mice.

The study addresses some very interesting issues and has potentially highly significant implications for the theoretical understanding and clinical management of Parkinson's Disease. The experiments addressing the effects of 6-OHDA on GCINs and the NVCC1 block in Parkinson's are convincing. The findings are very novel and if further confirmed will significantly address the understanding of the basal ganglia.

Critique:

Major comment.

One of the key assertions of the manuscript is that the Lhx6 co-expressing CINs (GCINs) are genuinely GABAergic. Although the authors provide convincing and clear evidence that 2 types of CINs exist in the striatum the GABAergic nature of the Lhx6+ cell-type is not convincingly demonstrated.

The conclusion that GCINs are GABAergic rests on 3 lines of results: (i) single-cell qRT-PCR shows expression of GAD65 in Lhx6/ChAT CINs, (ii) focal optogenetic stimulation aimed at ChR2 expressing genetically identified Lhx6+ GCINs elicit monosynaptic (nicotinic blockade insensitive) postsynaptic GABAergic IPSCs (as well as nicotinic EPSPs) in other CINs (iii) and in 1 of 29 tested pairs of GCINs presynaptic action potentials elicited dual nicotinic/GABAergic postsynaptic responses. Unfortunately the strength of these lines of evidence is insufficient to support these potentially very important conclusions. First, single-cell RT-PCR is notorious for false positive results, particularly in the slice preparation, and which is the reason why RT-PCR controls are characterized when using this technique. The manuscript states that "...Appropriate controls for possible DNA and RNA contamination were performed when harvesting cells, Reverse Transcription and qPCR reaction", but none of these results are presented either in figures or in numerical summaries in the text. Particularly troublesome is the fact that RT-PCR tests from Lhx6- CINs are not reported. If the 2 distinct CIN types really correspond to GABAergic and non-GABAergic neurons this should be a key experiment, not just as a methodological control but as a confirmation of the proposed classification scheme. The seeming low mRNA copy number for GAD65 (in GCINs) that can be inferred from the presented results together with the absence of immunocytochemical evidence for GAD65 in any ChAT positive cells either in this study or in the literature questions the generalization from the RT-PCR results that the Lhx6+ subtype is truly GABAergic. Although immunocytochemistry for GAD65 sometimes fails to identify genuinely GABAergic cells, if this concern is the underlying reason for not testing GAD65 with this method the authors could have employed a double transgenic strategy using GAD-Cre or VGAT-Cre lines. It would also be important if direct gel electrophoretic (or other) evidence was presented as positive confirmation of the sequence identity of the GAD65 amplicons. Second, the validity of the focal optogenetic stimulation experiments rests on the assumption that in the ChAT-ChR2-YFP line crossed with the Lhx6 reporter retains exclusive expression of ChR2 in cholinergic neurons. Recent experiment of several laboratories shows that the cell type specificity of transgene expression can be compromised even in well-characterized transgenic strains when they are crossed with other transgenics or a novel background strain. This raises the possibility that the ChR2-YFP+ cell population may include GABAergic cells or GABAergic afferents to the striatum from distal sources (PPN, GP ?) that are normally not cholinergic but acquire ectopic expression of ChR2-YFP in the double transgenic line. Since the focal light stimulation method cannot ensure that only the illuminated cell is activated (without sometimes recruiting axons nearby) the monosynaptic

IPSCs recorded may originate from non-cholinergic inputs.

Finally, the authors describe, but provide no graphical illustration of one paired-recording experiment in which monosynaptic nicotinic/GABAergic postsynaptic response could be elicited from a putative GCIN in a postsynaptic cholinergic neuron. Considering the problems with the above lines of evidence it is of key importance that the identity of the recorded presynaptic neuron is clearly demonstrated, however, the validity of this assertion cannot be evaluated without graphical illustration. We also think that although this is clearly demonstrated in the manuscript, the existence of GABAergic signaling by GCINs in 6-OHDA lesioned mice is not a valid line of argument for the existence of the same type of signaling in normal animals.

These concerns are further exacerbated by the failure of previous studies (Sullivan et al. 2008; English et al., 2012) to find any evidence for monosynaptic GABAergic responses (responses that could not be blocked by nicotinic antagonists) using (non-transgenic) rats or ChAT-Cre transgenic mice that exhibit very high efficacy of viral mediated targeting of ChR2 to ChAT+ interneurons.

In summary, the authors need to provide more direct evidence that GCINs are genuinely GABAergic in the normal striatum. Alternatively, limiting the conclusion to GABAergic signaling by these cells after 6-OHDA lesioning would also alleviate my concerns. The most convincing (although admittedly somewhat unrealistically difficult) experiment would be to obtain paired-recording evidence in which the presynaptic neuron is identified as cholinergic using ChAT immunocytochemistry and the IPSC is shown to be GABAergic and insensitive to the nicotinic antagonist cocktail.

Minor comments:

1. The decay time constant of the polysynaptic IPSC in this study appears to be much longer than in previous experiments. In fact, the reported time constant is similar to GABA_A-slow. What may be the reason for this ?
2. The authors should specify the temperature at which in vitro recordings were made.

Marseille, the 18th January

Dear Editors

Thanks for this excellent reviewing process with referees raising issues that are fundamental and to which we have replied and incorporated in the revised MS. We are grateful to the referees who clearly stressed the novelty and important implications for the understanding of basal ganglia in health and disease. The fundamental issues that referees and you have raised have been correctly dealt with new experiments:

- i) Direct immuno-cytochemical evidence of cholinergic-GABAergic interneurons in the dorsal striatum using GABA and ACh markers. Using VGAT and ChAT antibodies, more than 50% of ChAT⁺ cells show clear double labeling for VGAT (CGINs) confirming the dual phenotype of striatal cholinergic interneurons in control condition, both in wt and Lhx6-iCre;RCE-EGFP mice. 95% of ChAT⁺/VGAT⁺ cells also express GFP⁺ (Lhx6) (new Fig. 2ab and new supplementary Fig. 6). These data are in a good agreement with our single-cell PCR data.
Concerning specificity of the ChAT-ChR2 lines and the discrepancy with earlier studies and notably the excellent study of Tepper and colleagues. This can readily be explained by: a) the fact that CGINs are rare (about 0.5% of striatal neurons) in control mice and have a very low probability of interconnections and thus can easily be omitted, b) input-specificity issue (in fact they examined cholinergic interneurons - SPN connections not CGINs-CGINs), c) the strong short-term depression of direct GABAergic currents in CGIN-CGIN pairs (we used almost 300 msec intervals between stimuli). We verified the specificity of YFP (ChR2) expression in ChAT⁺ interneurons from ChAT-ChR2-YFP (98% of co-expression) and ChAT-ChR2-YFPxLhx6-iCre; AI14-tomato mice (95% of co-expression) (new supplementary Fig. 8, supplementary table 1).
- ii) The chloride issue –most fundamental. We have succeeded to record the activity of single GABA_A channels to determine DF GABA in striatal cholinergic neurons –done for the first time to the best of our knowledge- and found that after 6-OHDA, chloride levels are high in CGINs in comparison to control CGINs, providing direct and compelling data on this central issue. We also performed single NMDA channel recordings to determine the genuine V_{Rest} and to exclude possible alterations of this central parameter in CGINs in dopamine deprived conditions (new Fig. 4a-f).
- iii) Concerning the validation of our conclusions in control mice, we show:
 - 1) Lhx6/ChAT co-staining in wt mice (supplementary Fig. 1b) and in Lhx6-iCre;RCE-EGFP mice (Fig. 1a)
 - 2) VGAT/ChAT co-staining in wt (new supplementary Fig. 6c) and Lhx6-iCre;RCE-EGFP (new Fig. 2b) mice.
 - 3) GAD65/67/ChAT co-staining in wt (new supplementary Fig. 6b) and Gad1-GFP (new Fig. 2a) mice.
 - 4) GAD 65 and Lhx6 mRNAs in Lhx7⁺ /ChAT⁺ cells by RT PCR data in control Lhx6-iCre;RCE-EGFP mice (Fig. 1c)

5) Morphological & electrophysiological identifications of a directly connected CGIN-CGIN pair (both pre- and post-synaptic cells have the usual cholinergic features) (new supplementary Fig. 10)

6) We increased the number of optogenetic experiments in ChAT-ChR2-EYFP mice alleviating the concerns on dual mutant mice (ectopic expression of ChR2 by ChAT⁺ interneurons) (new supplementary Fig. 8b). We also verified the specificity of YFP (ChR2) expression in ChAT⁺ interneurons from ChAT-ChR2-YFP (98% of co-expression) and ChAT-ChR2-YFPxLhx6-iCre; AI14-tomato mice (95% of co-expression) (new supplementary Fig. 8a, supplementary table 1 and see also figure below).

Below you will find point-by-point response to the Reviewers' comments.

Reviewer #1 (Remarks to the Author):

The manuscript by Lozovaya et al. described a novel finding that many striatal cholinergic interneurons co-expressed GAD, and were capable of releasing both ACh and GABA. In addition, such co-transmission is impaired in a mouse model of Parkinson's disease (6-OHDA injection model). Mechanistically, the Cl transporter function is involved. The finding is novel and very interesting. The data are presented with high quality, and convincing. Therefore, we recommend publication in the journal.

Thanks a lot for these evaluation of the study and its implications, we have done our best to satisfy the requests of the referee in order to fully convince the readers of these implications.

However, we do have a few concerns and suggestions:

1. The previous work using ChAT-ires-Cre mouse and optogenetic tools only revealed di-synaptic inhibition, most likely by recruiting putative GABAergic neurons. Here, the authors used ChAT-ChR2-

YFP transgenic mouse, and *lhx6-iCre* mouse. To better integrate with existing literature, I would recommend the authors to perform characterization using similar approach shown in Fig 2., but instead, using *ChAT-ires-Cre;Ai14* mouse combined with staining for GABAergic neuron markers. This would solve the discrepancy. Potentially *ChAT-ChR2* mouse and *ChAT-ires-Cre* mouse may label different subsets of cholinergic interneurons.

This is truly fundamental and we are grateful to the referee for raising this issue. Indeed, there is in the literature a blind conviction on transgenic mice often without validating the underlying genetic labels. We have used several approaches to validate our conclusions including:

*i) VGAT and ChAT co-staining in *Lhx6-iCre;RCE-EGFP* and in wt mice (59 % of ChAT+ cells are VGAT+)(new Fig. 2b and new supplementary Fig. 6c).*

GAD65/67 labeling confirmed that somata of some cholinergic neurons are immunoreactive for GAD (new Fig. 2a and new supplementary Fig. 6b) but GAD immunoreactivity, due to noisy background, did not allow quantification (because it did not label cell bodies as well as VGAT).

*Concerning specificity of the *ChAT-ChR2* lines and the discrepancy with earlier studies and notably the excellent study of Tepper and colleagues. This can readily be explained by: a) the fact that CGINs are rare (about 0.5% of striatal neurons) in control mice and have a very low probability of interconnections and thus can easily be omitted, b) input-specificity issue (in fact they examined cholinergic interneurons - SPN connections not CGINs-CGINs), c) the strong short-term depression of direct GABAergic currents in CGIN-CGIN pairs (we used almost 300 msec intervals between stimuli). We also verified the specificity of YFP (*ChR2*) expression in *ChAT+* interneurons from *ChAT-ChR2-YFP* (98% of co-expression) and *ChAT-ChR2-YFPxLhx6-iCre; Ai14-tomato* mice (95% of co-expression) (new supplementary Fig. 8a, supplementary table 1 and see also figure below).*

The vast majority (95%) of *ChAT*⁺ express YFP in the dorsolateral striatum of *ChAT-ChR2-EYFPxLhx6-iCre; Ai14-tomato* mice.

YFP (green) and ChAT (red) co-immunostaining of coronal slices from *ChAT-ChR2-EYFPxLhx6-iCre; Ai14-tomato* mice showing co-expression of ChAT and ChR2-YFP.

A negative control without ChAT primary antibody was made because of cross-reactivity of the donkey anti-goat ALEXA-633 secondary antibody with cortico-striatal fibers.

Scale bar: 40 μ m.

2. Previous studies from Paul Bolam, and Jim Surmeier groups have shown that striatal cholinergic interneurons receive large amount of thalamic inputs. Here, the authors used cortical stimulation. It is worthy discussing whether two different cholinergic interneurons have different afferent inputs (cortical vs thalamic).

We agree this is a distinct possibility. In the absence of specific data on the referee's question and due to possible antidromic activation of thalamic fibers, we now added a sentence in the discussion page 10 line 25: "Whether, the pause response is 25 generated by the cortex or the thalamus as suggested by Surmeier and colleagues 1 cannot be 26 ascertained at present, but our results are in full accord with this study in stressing the crucial role of CGINs in the synchronization of striatal networks in relation to sensory integration".

3. When using low Cl and high Cl internals, the reported values in mV may be complicated by their different junction potentials. How much were the junction potential when using low and high Cl internals? Was junction potential corrected in the reported values?

Thanks, this is now specified. Liquid junction potentials were calculated using Clampex's Junction Potential Calculator (16 mV in "low Cl" solution vs 12.4 mV in "high Cl" solution, $\Delta 3.6$ mV) and corrected by the Pipette Offset circuitry of the amplifier (See Methods, page 13, lines 23-35).

Reviewer #2 (Remarks to the Author):

The manuscript by Lozovaya et al. advances two very interesting ideas. First, they argue from lineage markers that there are two populations of striatal cholinergic interneuron: one that releases acetylcholine alone and another that co-releases GABA. In addition, they differ in dendritic anatomy and possibly physiology, although that is not as clear. Second, the authors argue that following lesioning of the dopaminergic innervation of the striatum, the intracellular chloride concentration rises specifically in interneurons that co-release GABA, causing GABAergic signaling to become less inhibitory, causing aberrant activity patterns and promoting parkinsonian behavior. Although the results are intriguing, only the first half of the paper is convincing. Indeed, there appear to be two populations of cholinergic interneuron. That said, the authors need to provide a better characterization of these two particularly with regard to their physiology and ability to co-release GABA. The data on this latter point is particularly problematic. Insofar as the second idea is concerned, the results are far from convincing that the chloride reversal potential changes in a subpopulation of interneurons and that this change is causally linked to the parkinsonian motor phenotype. These and other concerns are outlined below.

We thank the Reviewer for extremely comprehensive review. Unfortunately, responding to all these demands would require numerous laboratories and teams for quite a few years. Nevertheless, we have made our best to respond to some of the wide range exhaustive list of the referee

Major concerns:

- Based upon the scRT-PCR data in Figure 1, there is no relationship between GAD65 expression and Lhx6 expression. This is not clear from reading the text. Wasn't this the starting hypothesis?

In single-cell PCR experiments, things are not simple, the material removed from every cell does not necessarily give all the mRNAs expected, at any rate the limitations of this technique are discussed. The absence of correlation between GAD65 and Lhx6 expression could be explained just by technical

issue, and difficulties to detect reliably Lhx6 signal because of extremely low amount of material. We now show with immunohistochemistry that 98% of YFP⁺ (Lhx6)/ ChAT⁺ interneurons are also positive for VGAT in wt and Lhx6-iCre;RCE-EGFP mice (new Fig. 2, new Supplementary Fig. 6b and Supplementary table 1).

- Was vGAT mRNA present? Previous work in other cell types has reported co-expression of ChAT, GAD and vGAT expression as I recall. What about NKCC1 mRNA?

We have now performed VGAT staining and confirmed dual labeling of CGINs. In particular 56% of ChAT⁺ cells were VGAT positive. Furthermore, 98% of YFP⁺ (Lhx6⁺) neurons were also VGAT⁺. These data are in a good agreement and confirm results obtained by iDisco and validate the link between Lhx6 and GABAergic neurons.

With due respect, we did not have the intention of testing all possible candidate markers. Showing NKCC1 would not add anything to our understanding as its presence neither indicates its levels of phosphorylation nor its activity that is relevant in the context of the polarity of GABA.

- Cholinergic interneurons have previously been shown to co-release glutamate and acetylcholine. Was vGluT3 co-expressed in either population?

Again, this paper is not aimed at determining whether these synapses are also glutamatergic. We observed CNQX-sensitive excitatory component in both cholinergic neurons and SPNs with optogenetic stimulation of cholinergic cells but did not comment on it since it is not the topic of the paper. In contrast in paired recordings the inward component was completely blocked by nicotinic antagonists.

- The evidence that cholinergic interneurons receive monosynaptic GABAergic input from other interneurons in the Lhx6+ class is not very convincing. Was the GABAergic response in interneurons evoked by optical stimulation of Lhx6+ interneurons sensitive to ionotropic glutamate receptor antagonists? A more direct and simpler test is to bath apply TTX (to block conducted activity) and micromolar 4-AP and then optogenetically stimulate again. If there are interneuron synapses on the cell being recorded from, you'll see the synaptic response.

Yes this is the case as CNQX does not block GABAergic signal (new Supplementary Fig. 8).

We have succeeded in performing that experiment with optogenetic stimulation of cholinergic cells in the presence of TTX and 4-aminopyridine. We recorded a direct GABAergic response in condition when all polysynaptic circuitries have been suppressed (new Supplementary Fig. 9).

- In previously published work, the spiking rate of cholinergic interneurons in ex vivo slices from 6-OHDA lesioned mice is not significantly different from that found in unlesioned mice.

Indeed, in the present work we found a difference of CGINs frequency between control and 6-OHDA mice. The discrepancy from previously published work can come from the fact that CGINs and CINs were mixed in these studies. Note that in control striatum, CINs frequency is much higher than that of CGINs (Fig. 1g). One of our main conclusions is that cholinergic interneurons are not homogeneous,

challenging many conclusions derived from recordings of ChAT positive neurons without discriminating their features.

- An increase in dendritic surface area is not strong evidence for an increase in collateral connectivity. More direct evidence is necessary. Also, was the change in dendritic surface area with lesioning peculiar to Lhx6+ interneurons?

An increase in dendritic tree suggests a bigger volume of targets for synapse formation; more dendrites imply more possibilities of connections. We have also some evidence of axonal increase, but the extraordinary volume of the axons precludes a detailed quantification. "More direct evidence" of increased connectivity is an increased probability of finding connected pairs in paired patch-clamp recordings. Respectfully, it might be important to stress to the referee the study of Gittis and colleagues showing dendritic sprouting of PV⁺ striatal neurons after 6-OHDA lesion (Neuron. 2011 Sep 8; 71(5): 858–868) where a similar approach led to a similar conclusion: there is little doubt that enhanced dendritic volume will facilitate the formation of more connections.

The second sentence is not relevant to the aims of this paper, surely, the referee does not imply that we must reconstruct all striatal neurons to see which ones sprout and which do not. However, our pilot data (related to another ongoing project) do show increased sprouting of PLTS dendrites.

- The striatal network is highly interconnected with many GABAergic neurons that will be sensitive to bumetanide and isoguvacine. This makes the derivative outcome measure of spontaneous spiking rate of questionable value to the conclusions about the Cl⁻ reversal potential. A direct assessment of this value in Lhx6+/- interneurons using perforated patch techniques must be made. This measure should be made in control and lesioned mice in both interneuron subtypes.

We have now performed this experiment and to the best of our knowledge for the first time in striatal ChAT+ neurons: single GABA_A channel recordings in control and dopamine-deprived CGINs. We found clear differences in DF GABA. As a matter of fact, perforated patch is not totally non-invasive (permeability for sodium can consequently distort chloride homeostasis through transporters). The single GABA channels experiments confirmed shift in DFGABA to more depolarized values and increase of intracellular chloride levels in CGINs after 6-OHDA lesion compared to the control condition (new Fig. 4a-f). According to these data, theoretically estimated increase of [Cl⁻] is about 10 mM. In addition, direct increase of intracellular chloride in one CGIN (shown in Fig 5) is sufficient to abolish the pause-response in this CGIN that allows excluding the involvement of any other striatal network players.

Experiments in CINs is technically very complicated by the facts of extremely low number of these cells, difficulties to find them "blindly" and they can be mixed up with CGINs due to "bleaching" of fluorescence.

- The cholinergic interneuron burst-pause pattern of responding to excitatory synaptic input has previously been shown to have several determinants. At higher burst frequencies and stronger stimulation intensities, GABAergic signaling is clearly one of these. So, the dependence of the pause

upon the intracellular Cl⁻ concentration seen here is expected. It is important to show that the pause evoked with this protocol is dependent upon GABA_A receptors.

We have now shown that the pause response and rebound in CGINs is blocked by GABA antagonist gabazine (new supplementary Fig. 14).

Is there any DA dependence to the pattern?

As we showed, 6-OHDA induced lesions led to nearly complete DA depletion in the dorsal striatum (checked by TH staining for all our slices from treated mice) and to the abolishment of the pause and rebound response (Fig. 5). Is the intention of the referee to require the determination of the actions of DA receptors on CINs and CGINs? This would require at least 1 more year work and is not mandatory to the main message of this paper.

Can the pause be self-induced by recurrent collaterals?

The pause is definitely not “self-induced by recurrent collaterals?” as the duration of GABAergic component of recurrent response is around 100-200 ms and undergo strong short-term depression and therefore completely terminated by the end of train stimulation. Most probably, pause is generated by activation of numerous surrounding interneurons recruited directly by cortical stimulation and via nicotinic receptors-mediated activation.

- Is there a change in NKCC1 expression in Lhx6⁺ interneurons following 6-OHDA lesioning?

There is no specific NKCC1 antibody and even if there would be, what would the intensity of immunocytochemistry told us on chloride removal? There is plenty of evidence that only the phosphorylated form is hyperactive and that is what matters.

- The work with the IPSPs in SPNs is a bit confusing. Previous work by the authors implicated PLTSIs – not cholinergic interneurons – as the source of these IPSPs in SPNs following 6-OHDA lesions. In fact, didn't this previous work claim they were insensitive to nAChR antagonists?

Indeed, in our previous paper (Dehorter et al 2009), we suggested that Giant GABAergic currents are most likely triggered by PLTS but not by cholinergic interneurons on the basis of the effect of one nicotinic blocker mecamylamine (5 μM).

Firstly, the present study does not contradict nor exclude that activation of PLTS secondary to CGINs activation (via nicotinic receptors, see Luo et al., <https://www.ncbi.nlm.nih.gov/pmc/articles/PMC3630781/>) and their involvement in the generation of giant IPSPs. Involvement of other GABAergic interneurons cannot be excluded.

Secondly, concerning the sensitivity of giant GABAergic currents to nicotinic antagonists we now report that giant IPSCs in SPNs are sensitive to a large cocktail of 3 different nicotinic receptor antagonists only (see Fig.6a, Methods) indicating that numerous different subtypes of nicotinic

receptors are involved; this explains the discrepancy with our previous paper, where only one antagonist was tested (mecamylamine).

- The behavioral experiments need to be more completely described. They cannot be reproduced from what is given. Also, the timing of drug administration needs to be clearly stated. The videos help but also raise concerns about the nature of the motor disability that is alleviated by bumetanide.

We have described in more details the behavioral tests and their evaluations. Timing of drug administration is added and experiments fully described (supplementary methods). It is not clear what are the nature of the concerns of the referee on the motor disability?

- Aside from the ambiguity in the methods, the behavioral effects of bumetanide in the 6-OHDA lesioned mouse are impressive. However, it is not reasonable to attribute those changes to high intracellular Cl⁻ concentration in Lhx6⁺ interneurons. Bumetanide will increase GABAergic inhibition of many neurons in the striatum (and the brain). With the availability of cell type specific manipulations using optogenetics or chemogenetics, there isn't any excuse for not keeping the evidentiary bar high. In fact, the chemogenetic approaches would even allow selective elevation in chloride conductance.

I am grateful to the referee to admit that effects are impressive but then the referee casts some doubt on whether bumetanide acts on these CGINs interneurons. Whether bumetanide also acts on other striatal neurons and other systems is most plausible (see discussion page 11, lines 6,7): "The additional contribution of a failure of inhibition in other interneurons- notably PLTS known to burst when dopamine deprived cannot be excluded".

The chemogenetic suggestion is largely theoretical and would require years of work to a large team, in addition to the fact that there is no, at present, a system that allows to infect CINs and CGINs separately.

- Lastly, the statistical approaches used for display and significance testing

We used the statistical tools used usually in these conditions and they are fully described in the paper (see supplementary materials, section Statistical analysis).

Minor concerns:

- The electrical stimulation protocol does not ensure that thalamo-striatal glutamatergic axons are not being stimulated. Not only is current spread an issue, but most thalamostriatal neurons have a cortical axon collateral. The authors should simply refer to the stimulation as capable of activating glutamatergic axons.

We agree this is a distinct possibility that is now added to Discussion in the reviewed MS in the discussion page 10 line 25 and following: "Whether, the pause response is 25 generated by the cortex or the thalamus as suggested by Surmeier and colleagues cannot be ascertained at present, but our results are in full accord with this study in stressing the crucial role of CGINs in the synchronization of striatal networks in relation to sensory integration".

- At what temperature were there recordings made?

Room temperature, now specified see methods page 13, line 16

- How were PLTSIs identified?

Explained in the revised MS, See methods page 14, lines 21-23: "In addition to fast spikes, low 21 threshold spiking (PLTS) interneurons displayed low-threshold spikes when depolarized from 22 potentials near -70 mV or after cessation of hyperpolarizing pulses (see Fig. 1b)".

- In figure 3, were AMPA and NMDA receptor blockers present?

We performed this experiment with CNQX (new supplementary Fig. 8).

Doesn't the similarity in the delays of the GABAA IPSP with optogenetic and electrical stimulation imply both are polysynaptic?

In Figure 3 we have shown that direct and polysynaptic responses display different delays after optogenetic stimulation (see Figure 3a and b).

Similarity of delays of the direct GABA_A IPSC induced by electrical and optogenetic stimulation (Fig. 3a and Fig. 3e,f) reflects direct stimulation of presynaptic cells in both cases (optogenetic and electrical).

- Fig. 1: Is the apparent absence of a rebound in Lhx6⁻ interneurons simply a reflection of the weak pause? Is this rebound intrinsically generated or synaptic? Is the evoked increase in spiking different? In the raw data (j) it appears so.

Indeed, it is true, that pause is often correlated with the appearance of rebound. However, in optogenetic experiments pause response was without rebound. This could imply either network phenomena or accumulation of chloride after chloride influx during massive and prolonged GABA inputs responsible for rebound in experiments with cortical stimulation. In optogenetic experiments chloride could be distorted by massive influx of sodium by raw stimulation, which in turn stimulate Na-Cl exchange and artificially decreased chloride concentration.

Evoked spiking within train in cell-attached mode was highly variable, but spiking patterns in the train were not different irrespective of the absence or presence of a pause.

- What controls were run for contamination? None are obvious in the records presented. mRNA for D1 dopamine receptors or substance P or enkephalin should be undetectable.

Following controls were run to avoid false positive results due to contamination:

- 1) RT PCR was performed in individual cells (not pulled data set, as typical practice for these studies) to avoid false positive result due to contamination.*
- 2) SPNs were used as negative control*
- 3) test of water instead of cell DNA in the qRT PCR mix to check contamination in the qRT PCR mix*
- 4) samples with no reverse transcriptase to check DNA contamination in the reverse transcription mix*
- 5) Pipette immersed in slice without patching and then same procedure to check contamination in the pipette solution and the medium of recording.*

The other requirements of the referee are aside our aims but note that due to limited amount of material in the sample, technically, only 4 primers could be used for analysis simultaneously. On the other hand, we did not find any markers which are not expressed in cholinergic cells, but enriched in other striatal cells.

The referee might be aware that the negative controls he (she) suggested are not relevant as they are expressed in cholinergic cells (see refs for D1 receptors:

<https://www.ncbi.nlm.nih.gov/pmc/articles/PMC4204445/>

for Substance P (neurokinin-1) receptor mRNA:

<https://www.ncbi.nlm.nih.gov/pubmed/1718557>

<https://www.ncbi.nlm.nih.gov/pubmed/1692261>

Nevertheless in our experiments MSNs were used as negative control.

- Figure 5p is not justified.

We now justified the figure 5p (new legend Fig. 5)

- In figure 4c, why was time window different from figure a and b?

Corrected

Reviewer #3 (Remarks to the Author):

The main findings of the study are: (i) that, striatal cholinergic interneurons (CINs) can be subdivided into 2 distinct subtypes that differ in terms of their morphology, intrinsic electrophysiological properties and postsynaptic responses (including a pause response) to activation of cortical inputs, (ii) that, one of these types, representing about 50% of all CINs also express the LIM homeobox protein Lhx6 and the GABAergic marker GAD65, (iii) the latter class of cells, thus termed GABAergic-CINs (GCINs), are GABAergic and cholinergic in terms of signaling properties (iv), 6-OHDA lesion of the nigrostriatal dopaminergic pathway increases the connectivity (GABAergic and cholinergic) of GCINs, dramatically changes the morphology of these cells and blocks the expression of the cortically evoked pause response due to abnormal intracellular chloride levels ([Cl]⁻_i), (v) the application of a NKCC1 chloride transporter antagonist (bumetanide) restores normal [Cl]⁻_i and the cortically evoked pause in GCINs, and finally, (vi) that, the same drug ameliorates symptoms of Parkinsonism in 6-OHDA lesioned mice.

The study addresses some very interesting issues and has potentially highly significant implications for the theoretical understanding and clinical management of Parkinson's Disease. The experiments

addressing the effects of 6-OHDA on GCINs and the NVCC1 block in Parkinson's are convincing. The findings are very novel and if further confirmed will significantly address the understanding of the basal ganglia.

Critique:

Major comment.

One of the key assertions of the manuscript is that the Lhx6 co-expressing CINs (GCINs) are genuinely GABAergic. Although the authors provide convincing and clear evidence that 2 types of CINs exist in the striatum the GABAergic nature of the Lhx6+ cell-type is not convincingly demonstrated.

The conclusion that GCINs are GABAergic rests on 3 lines of results: (i) single-cell qRT-PCR shows expression of GAD65 in Lhx6/ChAT CINs, (ii) focal optogenetic stimulation aimed at ChR2 expressing genetically identified Lhx6+ GCINs elicits monosynaptic (nicotinic blockade insensitive) postsynaptic GABA_A IPSCs (as well as nicotinic EPSPs) in other CINs (iii) and in 1 of 29 tested pairs of GCINs presynaptic action potentials elicited dual nicotinic/GABAergic postsynaptic responses.

Unfortunately the strength of these lines of evidence is insufficient to support these potentially very important conclusions. First, single-cell RT-PCR is notorious for false positive results, particularly in the slice preparation, and which is the reason why RT-PCR controls are characterized when using this technique.

The manuscript states that "...Appropriate controls for possible DNA and RNA contamination were performed when harvesting cells, Reverse Transcription and qPCR reaction", but none of these results are presented either in figures or in numerical summaries in the text. Particularly troublesome is the fact that RT-PCR tests from Lhx6- CINs are not reported. If the 2 distinct CIN types really correspond to GABAergic and non-GABAergic neurons this should be a key experiment, not just as a methodological control but as a confirmation of the proposed classification scheme.

We are grateful to the referee for his (her) excellent remarks and have performed many additional experiments. The following controls were run to avoid false positive results due to contamination (added in supplementary material, Appendix 1):

- 1) RT PCR was performed in individual cells (not pulled data set, as typical practice for these studies) to avoid false positive result due to contamination.*
- 2) SPNs were used as negative control*
- 3) Water was tested instead of cell DNA in the qRT PCR mix to check contamination in the qRT PCR mix*
- 4) samples with no reverse transcriptase to check DNA contamination in the reverse transcription mix*
- 5) Pipette immersed in slice without patching and then same procedure to check contamination in the pipette solution and the medium of recording.*

Concerning PCR in Lhx6- CINs, unfortunately experiments in CIN is technically very complicated due to the extremely low number of these cells (just 0.5% of cells), difficulties to find them "blindly" and the probability to mix them with CGINs due to "bleaching" of fluorescence.

The seeming low mRNA copy number for GAD65 (in GCINs) that can be inferred from the presented results together with the absence of immunocytochemical evidence for GAD65 in any ChAT positive cells either in this study or in the literature questions the generalization from the RT-PCR results that

the Lhx6+ subtype is truly GABAergic. Although immunocytochemistry for GAD65 sometimes fails to identify genuinely GABAergic cells, if this concern is the underlying reason for not testing GAD65 with this method, the authors could have employed a double transgenic strategy using GAD-Cre or VGAT-Cre lines.

We have now performed VGAT /ChAT dual labeling in control mice and obtained direct compelling evidence that CGINs are dual transmitters expressing also GABAergic markers. We checked their occurrence using ChAT, GAD65/67 or VGAT staining in wt and Lhx6-iCre;RCE-EGFP mice.

- 1) *Lhx6/ChAT co-staining in wt mice (supplementary Fig. 1b) and in Lhx6-iCre;RCE-EGFP mice (Fig. 1a)*
- 2) *VGAT/ChAT co-staining in wt (new supplementary Fig. 6c) and Lhx6-iCre;RCE-EGFP (new Fig. 2b) mice.*
- 3) *GAD65/67/ChAT co-staining in wt (new supplementary Fig. 6b) and Gad1-GFP (new Fig. 2a) mice.*
- 4) *GAD 65 and Lhx6 mRNAs in Lhx7+ /ChAT+ cells by RT PCR data in control Lhx6-iCre;RCE-EGFP mice (Fig. 1c)*
- 5) *Morphological & electrophysiological identifications of a directly connected CGIN-CGIN pair (both pre- and post-synaptic cells have the usual cholinergic features) (new supplementary Fig. 10)*
- 6) *We increased the number of optogenetic experiments in ChAT-ChR2-EYFP mice alleviating the concerns on dual mutant mice (ectopic expression of ChR2 by ChAT⁻ interneurons) (new supplementary Fig. 8b). We also verified the specificity of YFP (ChR2) expression in ChAT⁺ interneurons from ChAT-ChR2-YFP (98% of co-expression) and ChAT-ChR2-YFPxLhx6-iCre; AI14-tomato mice (95% of co-expression) (new supplementary Fig. 8a, supplementary table 1 and see also figure below).*

The vast majority of ChAT⁺ (95%) express YFP in the dorsolateral striatum of ChAT-ChR2-EYFPxLhx6-iCre; AI14-tomato mice.

YFP (green) and ChAT (red) co-immunostaining of coronal slices from ChAT-ChR2-EYFPxLhx6-iCre; AI14-tomato mice showing co-expression of ChAT and ChR2-YFP.

A negative control without ChAT primary antibody was made because of cross-reactivity of the donkey anti-goat ALEXA-633 secondary antibody with cortico-striatal fibers.

Scale bar: 40 μ m.

It would also be important if direct gel electrophoretic (or other) evidence was presented as positive confirmation of the sequence identity of the GAD65 amplicons.

In this study we used microfluidic single cell real-time PCR. This technique is based on the detailed study “Microfluidic single cell real-time PCR for comparative analysis of gene expression patterns” Mary et al., Nature Protocols, 2012; with the only difference in the use of 96-wells plate, but not a Dynamic Array IFC. Results can be obtained from the Fluid Real-time PCR analysis software in three different formats: (a) results table, (b) image view, and (c) heatmap. Example of amplification plots for ChAT, Lhx6, Lhx7, GAD65 and HPRT mRNAs for a representative single EGFP+ cholinergic interneuron is shown in Supplementary Fig 2.

Second, the validity of the focal optogenetic stimulation experiments rests on the assumption that in the ChAT-ChR2-YFP line crossed with the Lhx6 reporter retains exclusive expression of ChR2 in cholinergic neurons. Recent experiment of several laboratories shows that the cell type specificity of transgene expression can be compromised even in well-characterized transgenic strains when they are crossed with other transgenics or a novel background strain. This raises the possibility that the ChR2-YFP+ cell population may include GABAergic cells or GABAergic afferents to the striatum from distal sources (PPN, GP ?) that are normally not cholinergic but acquire ectopic expression of ChR2-YFP in the double transgenic line. Since the focal light stimulation method cannot ensure that only the illuminated cell is activated (without sometimes recruiting axons nearby) the monosynaptic IPSCs recorded may originate from non-cholinergic inputs.

We have performed new experiments in ChAT-ChR2-EYFP mice. As indicated in results, optogenetic experiments are pooled data obtained in both ChAT-ChR2-EYFP mice (6 mice) and dual mutant ChAT-ChR2-EYFPxLhx6-iCre; AI14-tomato (2 mice). In both cases we obtained direct GABA-IPSC. Now we increased statistics for pure ChAT-ChR2-EYFP line mice (new Supplementary Fig 8). In addition, new experiments with 4AP and TTX exclude polysynaptic circuits in the generation of GABA-IPSCs (new Supplementary Fig. 9)

We also checked the specificity of YFP expression in ChAT+ interneurons from ChAT-ChR2-EYFP mice (98% of co-expression) and ChAT-ChR2-EYFPxLhx6-iCre; AI14-tomato mice (95% of co-expression). (new supplementary Fig. 8, supplementary table1 see also Figure below)

Scale bar: 40 μ m.

Finally, the authors describe, but provide no graphical illustration of one paired-recording experiment in which monosynaptic nicotinic/GABAergic postsynaptic response could be elicited from a putative GCIN in a postsynaptic cholinergic neuron. Considering the problems with the above lines of evidence it is of key importance that the identity of the recorded presynaptic neuron is clearly demonstrated, however, the validity of this assertion cannot be evaluated without graphical illustration. We also think that although this is clearly demonstrated in the manuscript, the existence of GABAergic signaling by GCINs in 6-OHDA lesions is not a valid line of argument for the existence of the same type of signaling in normal animals.

Thanks; an image of biocytin labeled presynaptic CGIN and Lucifer Yellow labeled postsynaptic CGIN are now shown with corresponding firing patterns and original traces unraveling their genuine cholinergic features (new Supplementary Fig. 10).

These concerns are further exacerbated by the failure of previous studies (Sullivan et al. 2008; English et al., 2012) to find any evidence for monosynaptic GABAergic responses (responses that could not be blocked by nicotinic antagonists) using (non-transgenic) rats or ChAT-Cre transgenic mice that exhibit very high efficacy of viral mediated targeting of ChR2 to ChAT+ interneurons.

Concerning specificity of the ChAT-Cre line and the discrepancy with earlier studies and notably the excellent study of Tepper and colleagues. This can readily be explained by: a) the fact that CGINs are seldom (about 0.5% of striatal neurons) and have a very low probability of interconnections and thus can easily be omitted, b) input-specificity issue (in fact they examined cholinergic interneurons-SPN connections not CGINs-CGINs), c) the strong short-term depression of direct GABAergic currents in CGIN-CGIN pairs (we used almost 300 msec intervals between stimuli).

In summary, the authors need to provide more direct evidence that CGINs are genuinely GABAergic in the normal striatum. Alternatively, limiting the conclusion to GABAergic signaling by these cells after 6-OHDA lesioning would also alleviate my concerns. The most convincing (although admittedly somewhat unrealistically difficult) experiment would be to obtain paired-recording evidence in which the presynaptic neuron is identified as cholinergic using ChAT immunocytochemistry and the IPSC is shown to be GABAergic and insensitive to the nicotinic antagonist cocktail.

Indeed this is ideal but not feasible unfortunately and this has to be admitted...yet we trust that the ensemble of data and approaches shows clearly that there are 2 populations of Chat neurons with one being mixed and its frequency is augmented in parkinson

Thanks for the suggestions and criticisms that led to new experiments. We trust that this evidence is now compellingly provided. The list of evidence of dual GABA/ACh interneurons in the dorsal striatum in control condition now includes:

- 1) Lhx6/ChAT co-staining in wt mice (supplementary Fig. 1b) and in Lhx6-iCre;RCE-EGFP mice (Fig. 1a)
- 2) VGAT/ChAT co-staining in wt (new supplementary Fig. 6c) and Lhx6-iCre;RCE-EGFP (new Fig. 2b) mice.

- 3) GAD65/67/ChAT co-staining in wt (new supplementary Fig. 6b) and Gad1-GFP (new Fig. 2a) mice.
- 4) GAD 65 and *lhx6* mRNAs in *Lhx7+* /ChAT+ cells by RT PCR data in control *lhx6-iCre;RCE-EGFP* mice (Fig. 1c)
- 5) Morphological & electrophysiological identifications of a directly connected CGIN-CGIN pair (both pre- and post-synaptic cells have the usual cholinergic features) (new supplementary Fig. 10)
- 6) We increased the number of optogenetic experiments in ChAT-ChR2-EYFP mice alleviating the concerns on dual mutant mice (ectopic expression of ChR2 by ChAT+ interneurons) (new supplementary Fig. 8b). We also verified the specificity of YFP (ChR2) expression in ChAT+ interneurons from ChAT-ChR2-YFP (98% of co-expression) and ChAT-ChR2-YFPx*Lhx6-iCre*; *Al14-tomato* mice (95% of co-expression) (new supplementary Fig. 8a, supplementary table 1 and see also figure above).

Minor comments:

1. The decay time constant of the polysynaptic IPSC in this study appears to be much longer than in previous experiments. In fact, the reported time constant is similar to GABA_A-slow. What may be the reason for this?

We did not quite get this question, because decay-time of polysynaptic IPSC depends on numerous factors. For example, amount of acetylcholine released upon stimulation and numbers and distance of interneurons recruited by acetylcholine diffusion in extracellular space, etc

2. The authors should specify the temperature at which in vitro recordings were made.

Room temperature now specified (Methods, page 13, line 16).

REVIEWERS' COMMENTS:

Reviewer #1 (Remarks to the Author):

The authors have sufficiently addressed all my major concerns. The data are of high quality. The results are interesting. I recommend this paper for publication.

Reviewer #2 (Remarks to the Author):

The authors have done a good job of addressing my primary concerns.

Reviewer #3 (Remarks to the Author):

My main concern with the initial submission of the manuscript was what I considered insufficient evidence for significant GABAergic inhibition by the GCIN subtype. The authors now provided satisfactory evidence to substantiate this novel and important finding (in particular the well documented paired recording, immunocytochemical evidence and controls for transgenics), and therefore I support publication of the manuscript. I request however that the authors mention in the discussion that the relative functional importance of GABAergic input to CINs from this source remains uncertain.